# Accurate imputation of human leukocyte antigens with CookHLA

Seungho Cook [1,2,11], Wanson Choi[1,11], Hyunjoon Lim [3,11], Yang Luo [4,5,6,7], Kunhee Kim[1,2], Xiaoming Jia[8], Soumya Raychaudhuri [4,5,6,7,9,10] & Buhm Han [1,2,3 ✉]

The recent development of imputation methods enabled the prediction of human leukocyte antigen (HLA) alleles from intergenic SNP data, allowing studies to fine-map HLA for immune phenotypes. Here we report an accurate HLA imputation method, CookHLA, which has superior imputation accuracy compared to previous methods. CookHLA differs from other approaches in that it locally embeds prediction markers into highly polymorphic exons to account for exonic variability, and in that it adaptively learns the genetic map within MHC from the data to facilitate imputation. Our benchmarking with real datasets shows that our method achieves high imputation accuracy in a wide range of scenarios, including situations where the reference panel is small or ethnically unmatched.

[1] Department of Biomedical Sciences, BK21 Plus Biomedical Science Project, Seoul National University College of Medicine, Seoul, South Korea. [2] Department of Convergence Medicine, University of Ulsan College of Medicine & Asan Institute for Life Sciences, Asan Medical Center, Seoul, South Korea. [3] Interdisciplinary Program for Bioengineering, Seoul National University, Seoul, South Korea. [4] Center for Data Sciences, Brigham and Women's Hospital and Harvard Medical School, Boston, MA, USA. [5] Division of Rheumatology, Inflammation, and Immunity, Department of Medicine, Brigham and Women's Hospital and Harvard Medical School, Boston, MA, USA. [6] Division of Genetics, Department of Medicine, Brigham and Women's Hospital and Harvard Medical School, Boston, MA, USA. [7] Program in Medical and Population Genetics, Broad Institute of MIT and Harvard, Cambridge, MA, USA. [8] Department of Neurology, University of California San Francisco, San Francisco, CA, USA. [9] Department of Biomedical Informatics, Harvard Medical School, Boston, MA, USA. [10] Centre for Genetics and Genomics Versus Arthritis, Manchester Academic Health Science Centre, University of Manchester, Manchester, UK. [11] These authors contributed equally: Seungho Cook, Wanson Choi, Hyunjoon Lim. ✉ email: buhm.han@snu.ac.kr

Human leukocyte antigen (HLA) genes in the major histocompatibility complex (MHC) region influence many immune-related disease phenotypes[1]. An essential step for understanding how the MHC complex affects disease susceptibilities is to fine-map the HLA alleles or amino acids that are driving disease associations[2]. Fine-mapping analyses of HLA often require thousands of individuals, because of the highly polymorphic nature of HLA and long-range linkage disequilibrium (LD) stretching across the MHC region. However, acquiring HLA information of individuals can be challenging, since neither the classical HLA typing nor next-generation sequencing[3] can easily scale-up to thousands of individuals given a cost limit. A popular alternative approach is to impute HLA from the SNP data of genome-wide association studies (GWAS), utilizing recently developed computational techniques specialized for HLA[4–6]. Because of their cost-effectiveness, these imputation approaches were widely adapted in HLA studies of many immune-mediated phenotypes, which led to the discovery of important amino acid positions and classical alleles[7–14].

Nevertheless, existing methods for HLA imputation still have limitations in imputation accuracy. The average error rates of previous methods range from 3 to 6% for predicting high-resolution (four-digit) HLA alleles[15]. Imputation of rare alleles is even more challenging because LD is weaker for rare alleles, the reference panel may have only a few copies of each allele, and the typed information can have ambiguity[16]. Another challenge is that the imputation accuracy depends on the reference panel. It is not always possible to have a large reference panel with a similar ethnicity as the target sample. Existing methods perform poorly if the reference panel is of low quality (different ethnicity) or low quantity (small size). If we could maximize accuracy with the same suboptimal reference panel, it may enhance HLA research in diverse populations. Moreover, existing methods often have feasibility problems. For example, the popular SNP2HLA becomes slow for samples larger than a ten thousand[5], and HIBAG can take a month to train with a large custom reference[6].

In this work, we present an accurate HLA imputation method called CookHLA. CookHLA substantially improves imputation accuracy over previous methods by implementing several changes. First, we employ the latest hidden Markov model[17,18] as our imputation engine. Second, we account for local variability in the highly polymorphic exons of HLA genes. We repeat imputation by embedding a marker set locally in each of the polymorphic exons and use consensus posterior probabilities from the repeated analyses. Third, we adaptively learn the genetic map of MHC from the data, which allows us to account for the data-specific and population-specific LD structure within MHC. We show that in the benchmarking with real datasets, CookHLA outperforms previous methods. For example, when we use the Type 1 Diabetes Genetics Consortium (T1DGC)[19] data as a reference panel and the HapMap data[20] as test data, our method reduces the imputation error rate of the predecessor method[5] by more than two-fold from 6.6 to 2.4%. Our method is shown to be more accurate in imputing rare alleles than other methods. Furthermore, our method shows a robust performance even with ethnically unmatched or small references, which suggests that our method can be useful in studying underrepresented populations.

## Results
**Overview of the method**. We developed an accurate HLA imputation method, CookHLA. Similar to its predecessor, SNP2HLA[5], CookHLA translates the multiallelic HLA information into a set of binary markers so that it can utilize an existing imputation algorithm (Fig. 1a). CookHLA employs the state-of-the-art imputation engine[18] that is superior to the one employed

by SNP2HLA. In addition, CookHLA uses two strategies to maximize imputation accuracy. As depicted in Fig. 1b, CookHLA repeats imputation while locally embedding prediction markers in each of the polymorphic exons and performs consensus calls. This strategy enables the binary markers to capture the local information contained in each exon more effectively compared to the naïve strategy of SNP2HLA that puts markers only in the center position of the gene. Next, CookHLA adaptively learns the genetic map from the data (Fig. 1c). Many imputation models, including the one we use, are based on the Li and Stephens model[21] that assumes each target individual as a mosaic of reference samples. In this model, the genetic map is used to determine how long a mosaic block stretches before switching to another block. Since the MHC region is notorious for the complex genetic structure that differs across populations[22], we can improve imputation by learning the population-specific and data-specific genetic map from data instead of using the widely used HapMap map obtained from averaging several populations[20].

**Imputation accuracy comparison**. Using various datasets, we benchmarked the prediction accuracies of differing methods. We measured the accuracy as a proportion of correctly predicted alleles at the P-group level ("Methods"). We compared our method CookHLA to three other methods: SNP2HLA[5], HIBAG-prefit, and HIBAG-fit[6]. HIBAG-prefit refers to running HIBAG with prefit parameters provided by the package by choosing an appropriate population. HIBAG-fit refers to fitting new parameters using a custom reference panel and then running HIBAG for prediction. For a fair comparison, we used the same reference panel for all methods, with the exception of HIBAG-prefit for which the parameters were already fit. Note that HIBAG-fit was not included in comparisons if the fitting was not completed in 1 month of computation.

We masked the HLA information in the target sample and imputed it to measure the accuracy. We first used the T1DGC data[19] ($N = 5225$) as the reference panel and the HapMap CEU data[20] ($N = 88$) as the target sample. The HapMap CEU data was typed for six HLA genes (HLA-A, -B, -C, -DRB1, -DQA1, and -DQB1), so we measured accuracy for these genes. When we ran SNP2HLA, the overall accuracy was 93.4%. In contrast, CookHLA achieved a much higher accuracy of 97.6% (Fig. 2a), reducing the error rate by more than half (6.6% vs 2.4%) using the same reference panel. We observed the largest accuracy gain in HLA-A. At this gene, CookHLA achieved 98.9% accuracy whereas SNP2HLA achieved 92.0%, thereby providing a sevenfold reduction in error rate (8.0% vs 1.1%). A large gain was also observed in HLA-DRB1, where CookHLA achieved 94.9% (error rate 5.1%), whereas SNP2HLA achieved 90.3% (error rate 9.7%). The improved imputation in HLA-DRB1 was encouraging because HLA-DRB1 is typically the most difficult to correctly impute[5], despite its key role in many autoimmune phenotypes[9,13,14]. For this benchmarking, we were not able to compare HIBAG-fit because the fitting of the T1DGC data did not finish in one month using 8 CPU cores (one core per each of eight genes in T1DGC). HIBAG-prefit was excluded as well because the prefit model included the HapMap CEU data in the reference panel[6].

Next, we tried imputing the EUR population in the 1000 Genomes data[23]. Since we examined accuracy for the HapMap CEU population in the previous analysis, we used the FIN, GBR, IBS, and TSI populations ($N = 404$) excluding the CEU population in this analysis. The 1000 Genomes database provides the HLA information of five genes (HLA-A, -B, -C, -DRB1, and -DQB1) inferred from the sequence data (see "Methods"). We used the T1DGC data ($N = 5225$) as the reference and EUR data as the target sample. The overall accuracy of SNP2HLA was

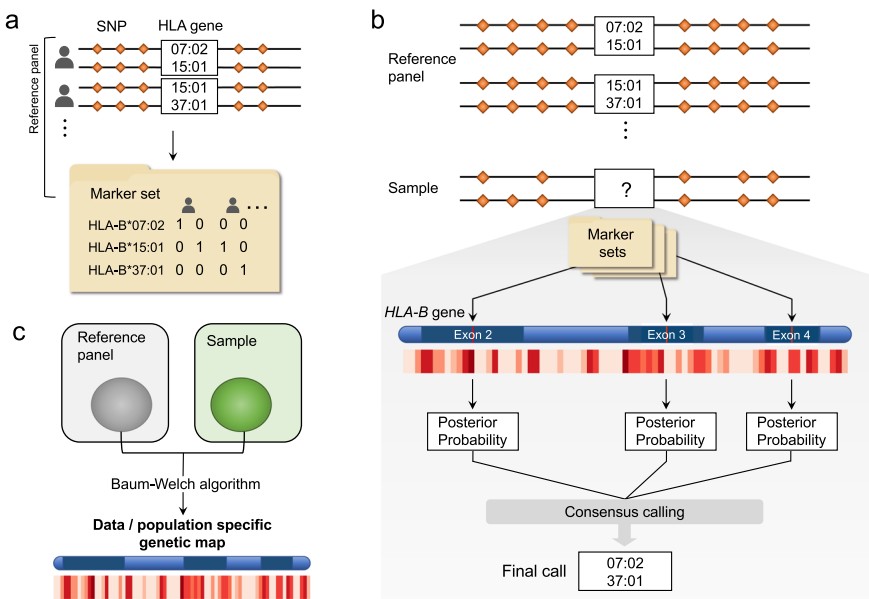

**Fig. 1 Overview of CookHLA. a** In the reference panel, CookHLA encodes HLA alleles as binary markers. **b** CookHLA repeats imputation while embedding a binary marker set into the middle position of each polymorphic exon. After the repeated imputations, CookHLA performs consensus calls by merging posterior probabilities. **c** CookHLA adaptively learns the data-specific genetic map within the MHC from the data.

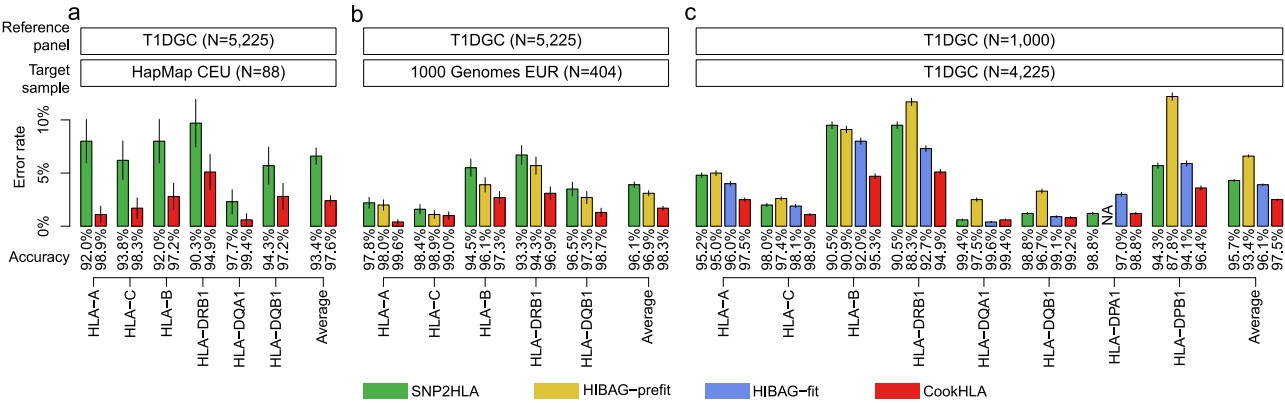

**Fig. 2 Imputation accuracy comparison.** Accuracies were measured based on the matches in the P-group level. **a** Prediction accuracy in imputing HLA of the HapMap CEU ($N = 88$) using the Type 1 Diabetes Genetic Consortium (T1DGC) data ($N = 5225$) as a reference. **b** Prediction accuracy in imputing the 1000 Genomes European (EUR) populations ($N = 404$) using the T1DGC data ($N = 5225$) as a reference. **c** Prediction accuracy in cross-validation using the split data from the T1DGC panel, where we used a subset as our target sample ($N = 4225$) and the rest as a reference ($N = 1000$). For HIBAG-prefit in (**b**) and (**c**), we used the European prefit model. The error bars represent SD.

96.1% (Fig. 2b). When we applied CookHLA, it achieved a higher accuracy of 98.3%. We also applied HIBAG-prefit using the European prefit parameters. The overall accuracy of HIBAG-prefit was 96.9%. Thus, CookHLA reduced the error rate nearly by half compared to HIBAG-prefit (3.1% vs 1.7%). However, this comparison might not be fair because the reference panel we used (T1DGC data) was different from the reference samples that HIBAG-prefit used. Similar to the previous analysis, HIBAG-fit was excluded due to the computational difficulty to fit the T1DGC panel.

In the third benchmarking, we simulated a smaller reference panel. To this end, we split the T1DGC data ($N = 5225$) into a relatively small reference ($N = 1000$) and target samples ($N = 4225$) and measured accuracy in eight genes (*HLA-A, -B, -C, -DRB1, -DQA1, -DQB1, -DPA1,* and *-DPB1*). For this reference size ($N = 1000$), HIBAG-fit finished the fitting of the model in one month (27 days), which allowed us to compare all methods in a controlled condition. In this benchmarking, the overall accuracy

of SNP2HLA was 95.7% (error rate 4.3%) (Fig. 2c). HIBAG-fit showed an improved accuracy of 96.1% (error rate 3.9%). When we applied CookHLA, we obtained a superior accuracy of 97.5% (error rate 2.5%). CookHLA was the most accurate in all HLA genes, except for *HLA-DQA1* where it was very close to the top method (99.4% compared to 99.6% in HIBAG-fit). In addition, we also applied HIBAG-prefit. The overall accuracy of HIBAG-prefit was lower than other methods (93.4%), mainly due to a low accuracy in *HLA-DPB1* (87.8%).

We also evaluated the accuracies of the methods in the Asian population dataset. Using the Chinese panel ($N = 9773$ after QC)[24] merged with the 1000 Genomes East Asian (EAS) population ($N = 504$)[23] as our reference, we imputed HLA in the Korean panel ($N = 413$)[25]. We used the Asian prefit model for HIBAG-prefit. HIBAG-fit was not feasible because of the large panel size. CookHLA achieved the best accuracy of 97.0%, while SNP2HLA achieved 91.1% accuracy and HIBAG-prefit achieved 94.6% accuracy (Supplementary Fig. 1).

**Decomposition of accuracy gain.** CookHLA differs from the predecessor, SNP2HLA, in that it uses (1) an upgraded hidden Markov model, (2) local embedding of markers on exons and consensus calling, and (3) adaptive learning of genetic map. We wanted to evaluate how much contribution each component provided on the accuracy gain of our method. To this end, we evaluated various versions of our method: (a) CookHLA with the old hidden Markov model (Beagle v3)[26], which is equivalent to SNP2HLA, (b) CookHLA with the upgraded model (Beagle v4)[17] but without a genetic map specified (when no map is specified, Beagle v4 assumes a map proportional to the physical position), which we call CookHLA-vanilla, (c) CookHLA with the upgraded model, no map specified, and with the local embedding strategy, which we call CookHLA-embed, (d) CookHLA with the upgraded model with the genetic map provided by HapMap[20] but without local embedding, which we call CookHLA-HapMap, (e) CookHLA with the upgraded model with the adaptive learning of genetic map but without local embedding, which we call CookHLA-adapt, and (f) the full CookHLA with the upgraded model, adaptive genetic map, and local embedding strategy. The full version is our method used for all previous analyses.

We used the T1DGC data[19] as a reference and the HapMap CEU data[20] as target samples similar to Fig. 2a. Supplementary Table 1 shows that all three components contributed to the final accuracy, where the use of the upgraded model contributed the most. Beginning from the bare version equivalent to SNP2HLA (93.47% accuracy), applying the upgraded model (CookHLA-vanilla) increased the accuracy to 97.00%. Then, applying additional strategies gradually increased the accuracy. When we applied the local embedding (CookHLA-embed), the accuracy slightly increased to 97.16%. When we applied the adaptive map (CookHLA-adapt), the accuracy also increased to 97.44%, which was better than using the HapMap map (CookHLA-HapMap; 97.06%). When we applied both the local embedding and adaptive map (full CookHLA), the accuracy was maximized as 97.63%. The combination of local embedding and adaptive map helped much in *HLA-DRB1*, where the use of the upgraded model only gave 93.18% accuracy and using all strategies gave 94.89% accuracy. Note that our final version (full CookHLA) was the most accurate in all genes.

This analysis showed that when we used a large reference panel of the same ethnicity, the upgraded model contributed the most and our two strategies contributed moderately. In the next section, we show that in a more difficult situation where there is no ethnically matched reference panel, our two strategies can help increase accuracy more significantly.

**Imputation using unmatched or small reference.** Although a large public reference panel is available for European[19] and East Asian[24], there are many populations for which large HLA panels have not been built. To study HLA in these populations using imputation, a possible strategy is to build a reference panel of the target population or to choose an existing reference panel that is as close as possible in ethnicity. However, a self-built reference panel will likely be small due to the cost limit, and an alternative panel may still show the subtle difference in ethnicity. If an imputation method can maximize accuracy on such a small or unmatched panel, it can have broad applicability in HLA studies of diverse populations.

In order to evaluate the performance of our method in situations where there is no ethnically matched large panel, we collected ten reference panels of different populations and sizes. These included the T1DGC panel (European, $N = 5225$), the 1958 Birth Cohort panel (European, $N = 918$), the Chinese panel (East Asian, $N = 9773$), the Korean panel (East Asian,

$N = 413$), the Pan-Asian panel (East and South Asian, $N = 530$), and the five populations of the 1000 Genomes data: AFR (African, $N = 661$), AMR (admixed American, $N = 347$), EAS (East Asian, $N = 504$), EUR (European, $N = 503$), and SAS (South Asian, $N = 489$). Then we considered every possible pair of these panels. For each possible pair, we evaluated the imputation performance by assigning one panel as a reference and another as a target. This comparison comprised a total of 90 test pairs ($10 \times 10 = 100$, excluding the 10 same-panel pairs). We then excluded the pair of Pan-Asian and the 1000G EAS in both directions due to their sample overlap, which led us to 88 test pairs.

Because this analysis required a lot of comparisons, running SNP2HLA (Beagle v3) was overly slow. Therefore, we instead made comparisons among CookHLA-vanilla, CookHLA-HapMap, and the full CookHLA, as defined in the previous section. CookHLA-vanilla can be considered as a direct update of SNP2HLA by upgrading the imputation engine. CookHLA-HapMap can be considered as a tuned version of CookHLA-vanilla with the genetic map of the HapMap. The performance gain of the full CookHLA versus CookHLA-vanilla can be seen as the lower bound of the performance gain of CookHLA versus SNP2HLA. The performance gain of the full CookHLA versus CookHLA-HapMap can be seen as a pure contribution of our two strategies: local exon embedding and adaptive map strategies. To further increase efficiency, we used Beagle v5 for this analysis instead of v4.

Table 1 shows the detailed results of the full CookHLA and CookHLA-HapMap in all 88 pairs (see Supplementary Table 2 for the results of CookHLA-vanilla). Figure 3 shows a scatter plot comparing the full CookHLA and CookHLA-HapMap (see Supplementary Fig. 2 for the scatter plots for CookHLA-vanilla). The full CookHLA almost always increased the accuracy over CookHLA-HapMap and CookHLA-vanilla, in 85 pairs out of 88. Even in the three pairs, the full CookHLA was very close to others.

Overall, the accuracy increase was observed in both ethnically matched and unmatched pairs (Fig. 3). The 88 test pairs consisted of 18 ethnically matched pairs (asterisks in Table 1) and 70 unmatched pairs. The average accuracies over the 18 matched pairs were 92.0%, 88.8%, and 88.1% in the full CookHLA, CookHLA-HapMap, and CookHLA-vanilla, respectively. Thus, the full CookHLA reduced the error rate by one-third (from 11.9% to 8.0%) compared to CookHLA-vanilla in the matched pairs. In some matched pairs, the accuracy of CookHLA-vanilla and CookHLA-HapMap were already high, for example when the T1DGC panel was used. For these pairs, the accuracy gain of the full CookHLA was relatively small. However, in other pairs with smaller reference sizes, the accuracy gain of the full CookHLA was more noticeable.

In the 70 ethnically unmatched pairs, the overall accuracies were lower than in matched pairs, as expected, because the ethnicities were different between reference and target. The average accuracies were 75.2%, 69.1%, and 68.8% in the full CookHLA, CookHLA-HapMap, and CookHLA-vanilla, respectively. Thus, the full CookHLA increased accuracy by 6.4% compared to CookHLA-vanilla. The largest accuracy increase was observed in the pair of Korean (reference) and 58BC (target). For this pair, the full CookHLA achieved 81.5% accuracy, whereas CookHLA-HapMap only achieved 64.3% accuracy.

In this analysis, there were two datasets for which an ethnically matched reference panel was not available: the 1000G AFR (African) and AMR (admixed American). For AFR, the best reference panel was the T1DGC (89.3% accuracy), and the second best was AMR (85.9%). For AMR, the best reference was the T1DGC (89.3%) and the second best was EUR (82.7%). Although

**Table 1 Pairwise accuracy benchmark using ten reference panels.**

| | | | Reference panel | | | | | | | | | |
|---|---|---|---|---|---|---|---|---|---|---|---|---|
| **Target sample** | Panel | T1DGC | 58BC | 1000G EUR | Chinese | Korean | 1000G EAS | Pan-Asian | 1000G SAS | 1000G AFR | 1000G AMR |
| | Population | EU | EU | EU | EA | EA | EA | EA, SA | SA | AF | AA |
| | Sample size | 5225 | 918 | 503 | 9773 | 413 | 504 | 530 | 489 | 661 | 347 |
| T1DGC (EU, 5225) | | | 91.3 (91.2)* | **93.3 (92.0)\*** | 80.4 (71.1) | 75.1 (58.6) | 73.0 (69.6) | 78.7 (73.3) | 85.1 (75.8) | 88.7 (76.5) | 91.0 (83.0) |
| 58BC (EU, 918) | | **97.5 (97.5)\*** | | 96.8 (95.4)* | 89.1 (83.1) | 81.5 (64.3) | 75.6 (63.8) | 74.1 (68.3) | 80.6 (73.9) | 92.0 (82.0) | 94.9 (87.6) |
| 1000G EUR (EU, 503) | | **98.3 (97.4)\*** | 95.8 (95.0)* | | 89.4 (87.4) | 81.0 (69.3) | 82.8 (75.9) | 76.0 (70.6) | 85.3 (79.4) | 91.1 (88.2) | 95.6 (87.7) |
| Chinese (EA, 9773) | | 80.8 (77.5) | 69.0 (63.8) | 73.2 (67.2) | | 90.5 (81.3)* | **92.3 (84.3)\*** | 89.2 (85.1)* | 80.3 (74.7) | 58.4 (52.7) | 68.4 (59.7) |
| Korean (EA, 413) | | 90.0 (87.8) | 71.9 (68.5) | 74.0 (65.9) | **95.4 (95.2)\*** | | 95.2 (86.0)* | 90.6 (85.8)* | 77.9 (70.4) | 61.9 (58.5) | 71.3 (58.5) |
| 1000G EAS (EA, 504) | | 91.3 (80.6) | 61.2 (48.3) | 68.8 (62.7) | **94.4 (94.4)\*** | 90.8 (82.0)* | Overlap | Overlap | 78.9 (77.8) | 57.9 (50.8) | 63.8 (60.4) |
| Pan-Asian (EA, SA, 530) | | **90.0 (89.9)** | 61.0 (61.1) | 64.3 (62.9) | 87.5 (89.7)* | 78.4 (73.4)* | Overlap | | 88.7 (82.3)* | 58.1 (54.9) | 59.9 (51.0) |
| 1000G SAS (SA, 489) | | **94.0 (90.8)** | 78.2 (74.3) | 82.9 (75.9) | 88.5 (85.6) | 79.5 (69.8) | 87.0 (83.6) | 83.8 (76.4)* | | 74.9 (63.3) | 79.4 (67.9) |
| 1000G AFR (AF, 661) | | 89.3 (84.8) | 68.9 (64.3) | 74.8 (68.8) | 59.1 (56.6) | 45.0 (37.8) | 50.2 (44.4) | 47.6 (42.8) | 53.3 (45.7) | | 85.9 (74.2) |
| 1000G AMR (AA, 347) | | 89.3 (88.7) | 78.9 (78.9) | 82.7 (79.3) | 77.0 (76.7) | 67.0 (57.8) | 68.7 (65.1) | 64.8 (59.0) | 72.4 (65.8) | 80.0 (77.3) | |

*T1DGC* Type 1 Diabetes Genetic Consortium, *58BC* 1958 Birth Cohort, *1000G* 1000 Genomes, *EU* European, *EA* East Asian, *SA* South Asian, *AF* African, *AA* admixed American. We collected ten different reference panels of various populations and sizes and tested each pair by assigning one as a reference and another as a target. The imputation accuracy was averaged over all available HLA genes for each pair. In each cell, the first number is the accuracy of the full version of CookHLA, while the number in the parentheses is the accuracy of CookHLA-HapMap (CookHLA with an only imputation engine upgrade, where the HapMap genetic map is used). The bold-faced font shows the highest accuracy for each target sample. Asterisks (*) indicate ethnically matched pairs. "Overlap" means that the pair was not tested due to sample overlap. For computational efficiency, we used Beagle v5 instead of v4 in this analysis.

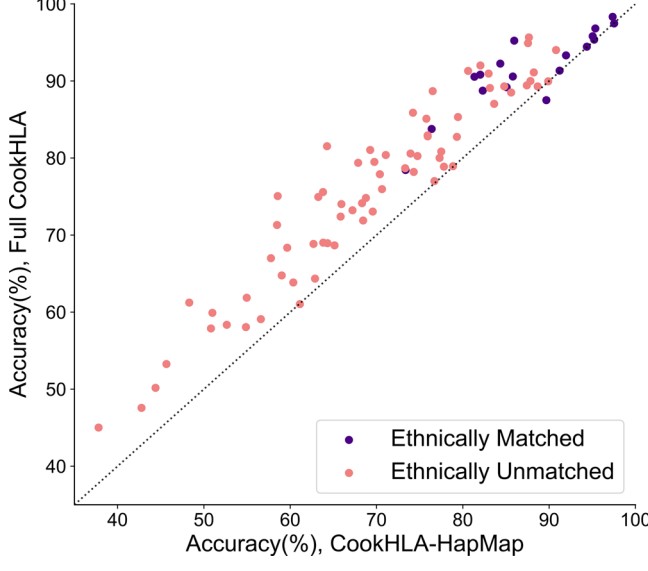

**Fig. 3 Pairwise comparison across ten different reference panels.** We collected ten reference panels of differing ethnicities and sizes. We then tested every possible pair by assigning one panel as a reference and another as a target, which comprised 88 tests after excluding overlapping sample pairs. We compared the full version of CookHLA to CookHLA-HapMap (upgraded engine, with HapMap genetic map). The dotted line indicates where the two methods' imputation accuracies are equal.

the T1DGC consistently showed good performance for these target populations, we note that our analysis might be biased because as described in "Methods", we confined the SNP set of the 1000G data to the Immunochip SNPs used in T1DGC due to computational efficiency.

**Imputation of rare alleles**. The IPD-IMGT/HLA database[27] shows that thousands of alleles exist for a single HLA gene. Naturally, many of the alleles are rare (≤1% population frequency). Some specific rare alleles have clinical implications, such as for disease susceptibility (HLA-DRB1*01:03 for Ulcerative colitis: 0.6% frequency)[7] or for adverse drug reaction (HLA-B*15:02 for reaction to carbamazepine, 0.3% frequency)[28]. Some known alleles are not strictly rare in our definition (>1%) but have low frequencies close to 1%: HLA-C*12:02 for late-onset psoriasis (1.1% frequency)[29], HLA-DRB1*08:01 for primary biliary cirrhosis (2.3% frequency)[30], HLA-B*57:01 for reaction to abacavir (1.7% frequency)[31], and HLA-B*58:01 for reaction to allopurinol (2.1% frequency; all frequencies of the alleles were estimated from the T1DGC panel)[32]. Moreover, rare alleles can be important for population genetic analyses[22]. Although we have used the average accuracy to evaluate methods thus far, the average accuracy tends to dominantly reflect the accuracy for common alleles. Therefore, it is worthwhile to evaluate the accuracy for rare alleles separately.

To this end, we measured the accuracy per each frequency bin. For each HLA gene, we categorized alleles into seven allele frequency bins: three rare allele bins (≤0.1%, 0.1~0.5%, 0.5~1%) and four additional bins (1–5%, 5–10%, 10–20%, and ≥20%). For each bin, we calculated the accuracies over the alleles in the bin using competing methods. Here we defined accuracy as sensitivity to correctly impute an allele given a true allele. We analyzed the cross-validation experiment of the T1DGC panel (Fig. 2c), for which all four methods were available.

Figure 4 shows that CookHLA is superior to other methods in imputing rare alleles correctly. In the lowest frequency bin (≤0.1%), every method had difficulties in imputing

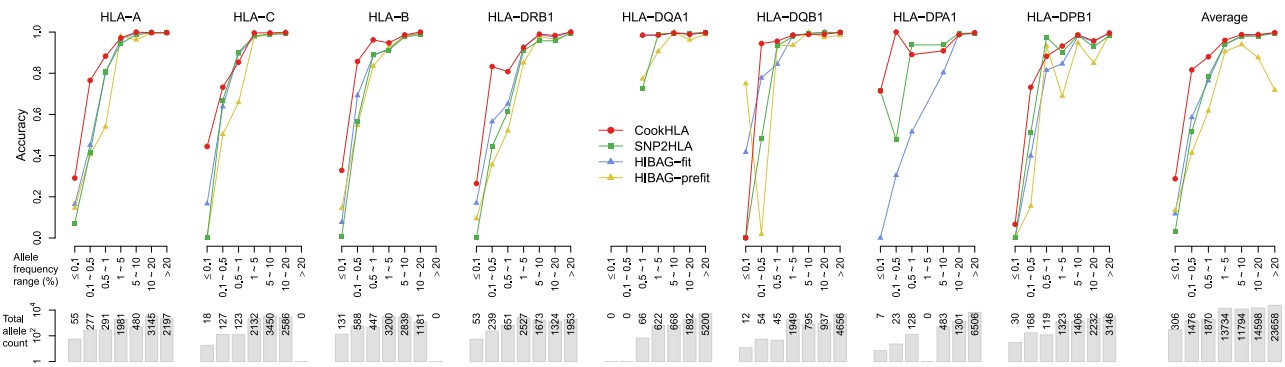

**Fig. 4 Imputation accuracy in each allele frequency bin.** We used cross-validation by splitting the T1DGC panel into reference ($N = 1000$) and the target sample ($N = 4225$). For HIBAG-prefit, we used the European prefit model. The accuracy refers to sensitivity to correctly impute an allele given a specific allele.

alleles correctly, but CookHLA achieved the highest accuracy (30.5%). The accuracy was nearly two times higher than the second best method (HIBAG-prefit:15.9%, HIBAG-fit:14.0%, SNP2HLA:3.2%). In the second lowest frequency bin (0.1–0.5%), CookHLA was the most accurate (80.0% accuracy), whereas the second best method's accuracy was only 56.6% (HIBAG-prefit:40.4%, HIBAG-fit:56.6%, SNP2HLA:50.4%). In the third lowest frequency bin (0.5–1.0%), CookHLA was, again, more accurate (92.3% accuracy) compared to other methods (HIBAG-prefit:71.0%, HIBAG-fit:81.9%, SNP2HLA:83.6%). In the bin of 1–5% frequency range, all methods showed ≥90% accuracy. In this bin, CookHLA was still the most accurate (95.6%), whereas the second best method's accuracy was 93.6% (HIBAG-fit). For higher frequency bins, the performances of the methods became comparable. For example, in the bin of 5–10%, CookHLA was the most accurate (98.8%), but the difference to the second best method (HIBAG-fit, 98.0%) was small compared to the difference observed in the lower frequency bins. The detailed accuracy estimates are in Supplementary Data 1.

We then wanted to compare different versions of CookHLA, as defined in the previous section. Supplementary Figure 3a shows that CookHLA-vanilla and CookHLA-HapMap achieved similar accuracy to the full CookHLA, except in (0.1–1.0%) bin *HLA-DRB1*, showing that the increased accuracy in rare alleles was mainly due to the imputation engine upgrade in this setting. However, when we repeated the similar analysis assuming the 1000 Genomes EAS data as reference and the Korean data as a target, the full CookHLA was much more accurate than CookHLA-vanilla or CookHLA-HapMap in lower frequency bins (Supplementary Fig. 3b). Thus, for the rare allele analysis, the contributions of the components in CookHLA showed a similar trend as were for the overall accuracy; for the situation where a large ethnically matched reference panel was used, the upgraded engine contributed the most, and for the situation where a small panel was used, the exon embedding and adaptive map strategies contributed significantly.

Since we defined accuracy as sensitivity, one possible concern in this analysis can be whether our method is overly imputing rare alleles. To examine this, in the cross-validation of T1DGC panel, we measured a positive predicted value (PPV) for each allele. Supplementary Fig. 4a shows that CookHLA has similar PPV as other methods. Finally, we calculated the F1-score, which is defined as the harmonic mean of sensitivity and PPV. When averaged over the genes, CookHLA showed a superior F1-score than other methods (Supplementary Fig. 4b). In the lowest frequency bin (0.1–0.5%), CookHLA achieved the highest F1-score (0.88) while the second best method HIBAG-fit achieved 0.78. In the second lowest frequency bin (0.5–1.0%), CookHLA

achieved the highest F1-score (0.89) while the second best method SNP2HLA achieved 0.83. In the third lowest frequency bin (1.0–5.0%), again, CookHLA achieved the highest F1-score (0.96) while the second best method HIBAG-fit achieved 0.94.

**Call rate and accuracy**. So far we have measured the accuracy of the methods assuming the best-guess imputed alleles. This corresponds to calling all alleles after imputation without considering the uncertainty. Sometimes, one may want to drop uncertain alleles and measure only the accuracy of the called alleles. Here, we analyzed the relationship between accuracy and call rate in different methods. We used the T1DGC-cross experiment in Fig. 2c. HIBAG-fit and HIBAG-prefit provide a confidence score for each genotype (pair of alleles). In contrast, CookHLA provides a consensus posterior probability for each allele. We defined CookHLA's confidence score for a genotype as the posterior probability of the allele for a homozygous call and the sum of the posterior probabilities of the two alleles for a heterozygous call. SNP2HLA does not explicitly provide a confidence score, but the posterior probability of each allele can be extracted from the output. Thus, we can build a similar score to CookHLA. Because the definitions of the scores were different among methods, we varied the score threshold for each method separately to measure accuracy versus call rate. Supplementary Fig. 5 shows that, as expected, the accuracy increased when the call rate was reduced in all methods. Decreasing the call rate did not change the relative performance of the methods; CookHLA was superior to other methods regardless of the call rate.

**HLA fine-mapping for three autoimmune diseases in WTCCC data**. We performed an example study of HLA fine-mapping using the data of the Wellcome Trust Case Control Consortium (WTCCC)[33]. HLA associations are common in autoimmune diseases. Therefore, among seven diseases of the original study[33], we used the three autoimmune diseases: rheumatoid arthritis (RA), T1D, and Crohn's disease (CD). We used the patients of diseases (RA, $N = 1860$; T1D, $N = 1963$; CD, $N = 1748$) along with 2938 controls. All samples were genotyped with the Affymetrix 500 K array chip. We imputed HLA of the samples using CookHLA and SNP2HLA with the T1DGC panel. We also used HIBAG-prefit using the European prefit model. We then examined the most significantly associated marker for each disease.

For RA, all three imputation methods gave an equivalent conclusion that the amino acid position 11 of HLA-DRβ1 was the most significantly associated with the disease (Fig. 5). The SNP at the second nucleotide of the codon for this amino acid, which codes for Val11 or Leu11, gave the most significant *p*-value

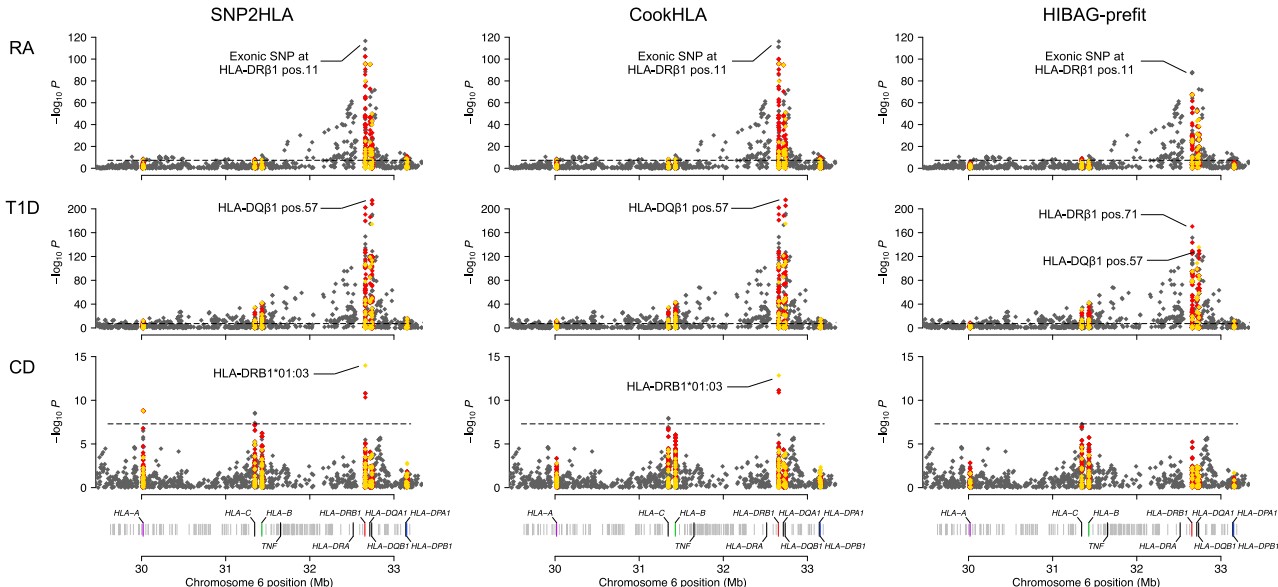

**Fig. 5 Association analysis of the WTCCC data.** We used different imputation methods to impute HLA of the WTCCC data and performed HLA fine-mapping analysis for three autoimmune diseases. RA rheumatoid arthritis, T1D Type 1 diabetes, CD Crohn's disease.

(CookHLA result: $OR = 2.87$ (2.62–3.14), $P < 10^{-116}$). This result was consistent with a large study that analyzed 19,992 samples, which found that the same SNP was the most significant ($OR = 3.7$, $P < 10^{-526}$)[9].

For T1D, CookHLA and SNP2HLA found the amino acid position 57 of HLA-DQβ1 to be the most significant (Fig. 5). The presence of alanine at this position conferred risk (CookHLA result: $OR = 4.95$ (4.48–5.47), $P < 10^{-215}$). This association is historically known[34] and was recently confirmed by a large study of 18,832 samples ($OR = 5.17$, $P < 1 \times 10^{-1089}$)[8]. HIBAG gave a slightly different result that the amino acid position 71 of HLA-DRβ1 was the top signal, where the presence of lysine conferred risk ($OR = 4.23$ (3.82–4.68), $P < 10^{-170}$). This association was also reported by the same study[8] as one of the primary signals ($OR = 4.70$, $P < 1 \times 10^{-935}$).

For CD, CookHLA and SNP2HLA gave the result that HLA-DRB1*01:03 was the most significantly associated with the disease (CookHLA result: $OR = 2.75$ (2.10–3.59), $P < 10^{-12}$) (Fig. 5). This allele is historically known for its association with CD[35]. This finding was also consistent with a large study of 54,674 samples, which found this allele to be the most significantly associated with CD ($OR = 2.51$, $P < 10^{-61}$)[7]. In the CookHLA imputation, this allele showed a case allele frequency of 4.12% and control allele frequency of 1.55% ($OR = 2.75$), of which the effect size was consistent with this large study[7] that reported 2.35% case frequency and 1.06% control frequency ($OR = 2.51$). However, HIBAG did not find this association (Fig. 5). It turned out that HIBAG imputed HLA-DRB1*01:03 for no samples (frequency 0% in $N = 8509$).

Overall, in this example fine-mapping analysis for three diseases, SNP2HLA and CookHLA gave similar fine-mapping results, while HIBAG often gave different results. For T1D, HIBAG found that the amino acid position 71 of HLA-DRβ1 ($P < 10^{-170}$) was much more significant than the historically known amino acid position 57 of HLA-DRβ1 ($P < 10^{-129}$). For CD, HIBAG imputed DRB1*01:03 for no samples. We conjectured a possibility that this difference came from the different reference panels used. The reference panel used for HIBAG-prefit model was smaller ($N = 2668$) than the T1DGC panel ($N = 5225$), which might have caused the missing of HLA-DRB1*01:03 in the panel. However, when we examined the allele frequency

reported in the HIBAG study (Supplementary Table S5 of Zheng et al.[6]), HLA-DRB1*01:03 existed in the reference panel with a similar frequency (0.73% frequency) to the T1DGC panel (0.65% frequency). Another possibility is the methodological difference. Because the imputation models differ, the results can differ even using the same reference panel. To test this hypothesis, we should fit HIBAG using the T1DGC panel. However, it was not possible because the fitting did not finish in 1 month with eight CPU cores. All association results of the WTCCC data are in Supplementary Data 2.

**Computation time.** We measured the computation time of the methods. We used a computer server with Intel Xeon 2.1Ghz CPU. We utilized 8 Gb memory per CPU core. Thus, when we used nine cores, we utilized 72 Gb memory. Note that memory size is not related to HIBAG's efficiency, since HIBAG consumes little memory in both fitting and running. We measured the time to impute the EUR data ($N = 503$) using the T1DGC panel ($N = 5225$) (Task 1) as well as the time to impute the Korean data ($n = 413$) using the Chinese panel ($N = 9773$) (Task 2).

Supplementary Table 3 shows the results. SNP2HLA ran fast for Task 1, taking 2.0 h. However, for Task 2 where the reference panel was roughly doubled, it became much slower and took 33.2 h. SNP2HLA cannot take advantage of multiple CPUs. CookHLA with Beagle 4 was slower than SNP2HLA for Task 1 (19.4 h) and faster than SNP2HLA for Task 2 (29.1 h). Since CookHLA performs ensemble learning from nine runs, it can be easily parallelized with multiple CPUs. With nine CPUs, CookHLA became much faster, taking 2.6 h for Task 1 and 4.5 h for Task 2. CookHLA with Beagle 5 was even faster. With a single CPU, it took 1.4 h for both Task 1 and 2, and with nine CPUs, it only took about 10 min for both tasks. Note that for CookHLA, an additional time of about 0.5 h was required for preparing an adaptive genetic map. HIBAG-prefit was fast, taking 34 min for Task 1 and 11 min for Task 2. Since HIBAG-prefit performs a separate imputation for each of the seven genes, it can be parallelized. With seven CPUs, it took less than 10 min for both tasks. Note that if one wants to use a custom reference panel, HIBAG requires a separate fitting step, which can take a long time. Fitting a subset of the T1DGC panel ($N = 1000$) took 27 days using eight CPUs (one CPU per gene). Fitting the whole

T1DGC panel did not finish in 1 month. Even when we assigned 8 cores per gene (64 cores), the fitting did not finish in two weeks.

## Discussion

We have developed an accurate HLA imputation method, CookHLA. We evaluated the relative performance of our approach compared to competing methods using a number of real-data-based analyses. Our method performed well when the reference panel was small or ethnically unmatched and was accurate in imputing rare alleles. Our method was computationally feasible in all tested conditions. Incorporating a large custom reference panel is straightforward and efficient in our method.

Our method can have utilities in many different situations. In the situation where a large ethnically matched reference panel is available, existing methods are already accurate. However, CookHLA can increase the accuracy even further. For example, in the T1DGC-cross experiment (Fig. 2c), SNP2HLA and HIBAG-fit were accurate (95.7% and 96.1%, respectively), but our method increased the accuracy even further to 97.5%. Although the absolute increase was small, the relative reduction of error rate was not negligible (42% and 36% reduction of error rate compared to the two methods). In view of the fact that traditional typing methods often have ambiguities and errors, the accuracy of 97.5% in the resolution of the P-group can be considered very high and even approaching the clinical level. In another situation where there is no ethnically matched large panel, our method can be even more useful. We have shown that our method can extract most of the information from the panel even if the panel is ethnically unmatched or small.

In HLA, many rare alleles are known for their critical roles for traits[7,28–32]. Thus, an accurate imputation of rare alleles can have implications in both academic researches and clinical applications. To date, many studies compared accuracies of the methods in terms of the overall average, which may not reflect the accuracy for rare alleles specifically[5,6,15]. When we narrowed the scope to a low-frequency range, the performances of the methods appeared to differ greatly (Fig. 4). Typically, rare alleles are more difficult to study than common alleles due to their rareness and low statistical power[36]. Ultimately, however, studies on rare alleles will be an essential step in the personal care of people with rare alleles. We expect that our method will be a useful tool for future studies of rare HLA alleles.

In our benchmarking, we compared our method to SNP2HLA and HIBAG. There was another HLA imputation method called HLA*IMP:02[4,37]. HLA*IMP:02 uses a haplotype graph model that considers HLA types and SNPs as edges, and the relationship between HLA types and SNPs as paths. We were unable to compare to this method because the software became proprietary and not available for public use. A previous study[15] reported that HLA*IMP:02 using an internal reference panel was slightly less accurate than SNP2HLA using the T1DGC panel.

HIBAG[6] is different from SNP2HLA[5] or CookHLA in that it has a separate model fitting step. We found that the model fitting step can take a long time for fitting a large reference panel. However, the imputation time after fitting was fast. Since HIBAG comes with prefit models for several populations, HIBAG can be a good choice if one wants to obtain a reasonable accuracy in a short computation time. However, if one wants to maximize the accuracy or wants to use a custom reference panel, our method can be the method of choice.

Since CookHLA uses phased reference data, the phasing quality in the reference data can be an important factor in accuracy. Many of the existing reference panels available for CookHLA are not trio data, and therefore there may be phasing errors. As for the imputation accuracy of the individual alleles, we observed that the potential phasing errors are not much problematic, because our accuracy tests were all based on reference panels with imperfect phasing. However, if we perform haplotype analysis, the phasing errors may affect the results more severely. In the future, it would be ideal to build population-specific reference panels by minimizing phasing errors with trio or family data.

Our method introduced two strategies to improve imputation, namely local embedding and adaptive genetic map. When we compared the contributions of the two strategies, we found that the contribution of the adaptive genetic map was greater (Supplementary Table 1). One reason for this might be because for many datasets we used, only exon 2 and 3 for class I and exon 2 for class II were typed. Thus, the alleles had ambiguity. Since the class II genes were typed based on a single exon, it is not surprising that local embedding did not much help. Since NGS-based typing is becoming prevalent, we expect that in the future, reference datasets with six-digit allele information will be available. Then, we will be able to evaluate the actual gain of local embedding using these datasets.

Recently, there were great improvements in computational algorithms to infer HLA from next-generation sequencing data[3,38–40]. These methods turned out to be highly accurate when the sequencing depth was sufficient[41]. In fact, some reference panels we used in our analyses were generated using the HLA information derived from the sequencing data[42]. These advances in technology indicate that SNP-based imputation algorithms will become obsolete if the cost of next-generation sequencing is reduced to the cost of microarray technology in the future. However, currently, large-scale studies commonly use GWAS chips due to their low cost, as in the UK biobank study[43]. Therefore, there is an ongoing and essential need to predict HLA alleles as accurately as possible based on the SNP data. We expect our method to help researchers decipher the role of HLA using extensive genome data.

## Methods

### CookHLA

*Imputation based on binary markers.* CookHLA was built upon the binary marker framework that SNP2HLA utilized[5]. In the context of SNP imputation, there have been active developments of imputation models[26,44,45]. In order to utilize these models in HLA imputation, SNP2HLA coded the presence and absence of each allele as a binary marker (Fig. 1a). Suppose that an HLA gene has $K$ different alleles, $a_1, a_2, \ldots, a_K$. We can define $K$ binary markers, $b_1, b_2, \ldots, b_K$, each corresponding to each allele. Given $N$ diploid individuals ($2N$ chromosomes), the element of $b_i$ at $j$th chromosome is

$$b_{i,j} = \begin{cases} 1 & \text{if } a_i \text{ is present at } j\text{th chromosome} \\ 0 & \text{if } a_i \text{ is absent at } j\text{th chromosome} \end{cases} \quad (1)$$

Given the reference samples with both HLA and intergenic SNP information, we generate these binary markers and combine them with the intergenic SNP data to build a reference panel. Then, suppose that we are given target samples to impute HLA, for which only intergenic SNP data are available. We can perform imputation using the reference panel to fill in the missing values at the binary markers, which allows us to predict the HLA alleles of the target samples. Note that since a recent imputation framework supports modeling of multialleles[18], an alternative approach would have been directly modeling alleles instead of using binary markers. In the current study, this alternative option was not explored.

**Upgraded imputation model.** To impute the binary markers, CookHLA employed a state-of-the-art imputation model (Beagle v4 and v5) that can deal with millions of samples[17,18]. Previously, SNP2HLA employed an old model (Beagle v3)[26]. While the old model is based on the variable-length hidden Markov model based on haplotype graphs, our model is based on the standard hidden Markov model that can incorporate the genetic distance as input. In particular, our model improved both efficiency and accuracy by including only the common markers between target samples and reference samples in the model and using interpolation for the rest of the markers[17], and by using a parsimonious state space that is a fraction of the size of the state space of the full model[18]. We found that Beagle v4 and Beagle v5 gave almost identical results in our application. In this study, we used Beagle v4 for most of the analyses, unless otherwise stated.

These new Beagle models had functionality that can incorporate multiple alleles at a locus. However, they were implemented to only accept A, T, C, and G or compositions of these letters as alleles, and thus were not directly applicable to HLA nomenclature. Therefore, we chose to use the existing scheme that defines one binary marker per each HLA allele.

Similar to SNP2HLA, CookHLA uses phased reference data and unphased SNP input data. When unphased data is provided as input on the new Beagle models, the data is first phased internally and then imputed. Although the internal procedure has changed, its use from the user's point of view is the same as with SNP2HLA. CookHLA uses the same phased reference data format as SNP2HLA. Thus, any reference panel built for SNP2HLA can be used for CookHLA. In addition, if anyone wants to build a new reference panel for CookHLA, the MakeReference module of SNP2HLA can be used.

**Local embedding of prediction markers into polymorphic exons.** CookHLA differs from SNP2HLA[5] in strategies using the binary prediction markers. In SNP2HLA, after one generates a new set of binary markers corresponding to alleles for an HLA gene (e.g., *HLA-B*), one places the marker set in the center position of the gene (e.g., chr6: 31,431,577(hg19) for *HLA-B*). This strategy, however, is not necessarily optimal. There exist multiple exons in HLA genes that are highly polymorphic and whose sequences can determine most of the alleles, such as the exons 2, 3, and 4 in the Class I and exons 2 and 3 in the Class II genes[27]. Because LD decays with distance, placing the prediction markers near the highly variable regions can help with imputation. However, it is unclear which exon to put the marker set in; whichever exon we choose, other exons will be overlooked. To address this challenge, we have developed a strategy that repeats imputation multiple times while embedding each marker set locally in the middle position of each polymorphic exon (Fig. 1b). Specifically, CookHLA repeats imputation for exon 2, 3, and 4 in Class I and exon 2 and 3 in Class II with an embedded marker set. To make final consensus calls over multiple imputation results, we combine the posterior probabilities of the binary markers as follows.

Let $a_1, a_2, \ldots, a_K$ be $K$ alleles at a locus. Let $L$ be the set of exons that we target. We use $L = \{2, 3, 4\}$ for the Class I genes and $\{2, 3\}$ for the Class II genes. Let $b_1^{(l)}, b_2^{(l)} \ldots, b_K^{(l)}$ be the binary markers that were embedded in the exon $l$. After imputation, we obtain the imputed markers $\widehat{b_1^{(l)}}, \widehat{b_2^{(l)}}, \ldots, \widehat{b_K^{(l)}}$, which indicate the posterior probabilities of "presence" of the $K$ alleles. Since we repeat imputation $|L|$ times, we want to combine these values to make a consensus call. Let $j'$ and $j''$ be the two chromosomes of a diploid individual. The imputation model (Beagle v4 or v5)[17,18] gives phased output, so one possible way would be to calculate the average posterior probability per chromosome, such that $h(j) = \mathrm{argmax}_{k \in \{1, \ldots, K\}} \frac{1}{|L|} \left( \sum_{l \in L} \widehat{b_{k,j}^{(l)}} \right)$.

Then, the predicted genotype would become $(a_{h(j')}, a_{h(j'')})$. However, this approach will be vulnerable to phasing errors between exons. Therefore, instead, we chose to calculate the average posterior probabilities for both chromosomes together:

$$p(k) = \frac{1}{2} \sum_{j \in \{j', j''\}} \frac{1}{|L|} \left( \sum_{l \in L} \widehat{b_{k,j}^{(l)}} \right) \quad (2)$$

We find $k_1$, the allele with the largest $p(k)$, and $k_2$, allele with the second largest $p(k)$. Then, we decide the unordered genotype of $(j', j'')$ as

$$\mathrm{Genotype}(j', j'') = \begin{cases} (a_{k_1}, a_{k_1}) & \text{if } p(k_1) \geq 2p(k_2) \\ (a_{k_1}, a_{k_2}) & \text{if } p(k_1) < 2p(k_2). \end{cases} \quad (3)$$

Our approach can be thought of as an ensemble approach[46] in the sense that we combine results from multiple runs targeting each exon. In addition, we found that it helps to expand the ensemble structure to account for multiple model parameters. We run imputation with differing parameters for sliding window in the hidden Markov model (overlap parameter (number of markers) $\in \{3000, 4000, 5000\}$ in Beagle v4[17], or overlap parameter (cM) $\in \{0.5, 1, 1.5\}$ along with window parameter = 5 cM in Beagle v5[18]). As a result, we run $3 \times 3 = 9$ imputations for the Class I and $2 \times 3 = 6$ imputations for the Class II genes and use the averaged posterior probabilities. Similarly, when we run CookHLA using more than one reference panel, we can merge the results by averaging the posterior probabilities.

**Adaptive learning of data-specific genetic map.** Another difference of CookHLA from previous methods is that it adaptively learns the genetic map of MHC from data (Fig. 1c). Using an appropriate genetic map can be important for accurate imputation. One possible approach is to use the publicly available MHC genetic map offered by HapMap[20]. However, the HapMap estimates of genetic distances represent averages from multiple populations. Therefore, no population-specific or data-specific LD structure of MHC is considered using the HapMap estimates.

We estimate the genetic map from the data as follows. The hidden Markov model for imputation considers each sample as a mosaic of the reference individuals[44]. In this model, the application of the Baum–Welch algorithm[47] can approximate the transition probability between markers, the probability that the reference individual in the mosaic changes to another. In the standard model for

imputation[44], the transition probability is defined

$$\tau_m = 1 - e^{-4N_e r_m / |H|} \quad (4)$$

where $N_e$ is the effective population size and $r_m$ is the genetic distance between markers. Thus, we can calculate the genetic map ($r_m$) given the transition probabilities ($\tau_m$). In our application, we assumed an effective population size of $N_e = 10^4$, which does not affect the results because it cancels out when we feed the genetic map to the imputation framework. Given the reference panel and the target samples to impute, we randomly select $Q$ individuals from the reference and $Q$ individuals from the target sample (Fig. 1c). Using these $2 \times Q$ individuals, we use the Baum–Welch algorithm implemented in MACH v1.0[45] to obtain the transition probabilities. MACH tutorial (http://csg.sph.umich.edu/abecasis/MACH/tour/imputation.html) recommends using 200~500 subsamples, so we chose $Q = 100$ ($2 \times Q = 200$). We found that increasing this number had little effect on the performance of CookHLA (Supplementary Data 3). We convert the transition probabilities to the genetic map using Eq. (4), which is used as input information to CookHLA.

We note that adaptive learning builds the genetic map of intergenic SNPs and not that of the binary markers, since the binary markers are missing in the target samples. When we embed the binary markers in the exon, we augment the map by defining the binary markers' genetic map. We first calculate the center position of the exon in terms of the genetic distance and place the markers at that position while specifying a very small genetic distance ($10^{-12}$ cM) between the binary markers. This serves to suppress an undesired transition between the binary markers in the hidden Markov model since binary markers are by definition exclusive (a chromosome cannot have two alleles).

**Existing HLA imputation approaches**

*SNP2HLA.* SNP2HLA is the predecessor approach of CookHLA and is widely used for HLA imputation[5]. Given the binary markers defined in Eq. (1), SNP2HLA places only one marker set in the center position of an HLA gene. Let $\widehat{g_1}, \widehat{g_2}, \ldots, \widehat{g_K}$ be the best-guess imputed binary markers for $K$ alleles. We let

$$h(j) = k \in \{1, \ldots, K\} \text{ such that } \widehat{g_{k,j}} = 1 \quad (5)$$

Then, given two chromosomes $j'$ and $j''$ of a diploid individual, the predicted genotype of SNP2HLA becomes $\{a_{h(j')}, a_{h(j'')}\}$. If there are imputation errors, it is possible that $h(j)$ has multiple values (collision error) or no value (no call error) for a chromosome. The imputation model in SNP2HLA (Beagle v3)[26] uses a variable-length hidden Markov model that does not require genetic map information as input.

*HIBAG.* HIBAG is another imputation method that uses attribute bagging[6], a technique that uses an ensemble classifier[46] from random subsets of features. HIBAG randomly selects individuals and intergenic SNPs from a training dataset to find the most accurate ensemble classifier for predicting HLA alleles. HIBAG provides prefit built-in classifier models of seven genes (*HLA-A, -B, -C, -DRB1, -DQA1, -DQB1,* and *-DPB1*) for European, Asian, Hispanic, and African ancestries, of which only the models but not the genotype data are available to the public. For the European prefit model, they used 2668 individuals including the HapMap CEU individuals[20].

HIBAG also allows users to fit a new model using custom reference data, which should be done per each gene. We found that this fitting can take a long time. We tried to fit eight classical genes in T1DGC data using eight computer CPU cores (number of classifiers = 100), but the fitting did not finish in 1 month (8 CPU months). We then confined SNPs to flanking regions (500 kb on each side of a gene) and tried 64 CPU cores (8 cores per gene), but the fitting did not finish in 2 weeks (32 CPU months). We used Intel Xeon 2.1Ghz CPU for this benchmark.

**Datasets**

*HapMap CEU panel.* HapMap CEU panel[48] includes 4638 SNPs within the MHC region (chr6:26-34 Mb) genotyped using the Illumina GoldenGate platform and 4-digit classical HLA types for *HLA-A, -B, -C, -DQA1, -DQB1,* and *-DRB1* from European individuals. HLA typing was carried out using PCR-SSOP-based protocols. For typing of class I, exon 2, intron 2, and exon 3 were examined. For typing of class II, exon 2 was examined. Genotype ambiguities were then resolved by direct sequencing of the whole PCR fragment. We downloaded the SNP2HLA-formatted data for 124 individuals from the SNP2HLA website[5], which included 88 unrelated individuals.

*Type 1 Diabetes Genetics Consortium panel.* The Type 1 Diabetes Genetics Consortium (T1DGC) panel[19] is the largest-to-date European reference panel for SNP2HLA. This dataset includes 5868 SNPs within the MHC region (chr6:29-34 Mb) genotyped using the Illumina Immunochip platform and 4-digit classical HLA types for *HLA-A, -B, -C, -DPA1, -DPB1, -DQA1, -DQB1,* and *-DRB1*. HLA was typed using the SSOP technology by performing PCR amplification of exon 2 and 3 in Class I and exon 2 in Class II[49]. We downloaded the SNP2HLA-formatted data for 5225 individuals from the SNP2HLA website[5]. This panel is available for research purposes per request.

*1000 Genomes panel*. The 1000 Genomes project[23] provides data for five super populations: African (AFR; $N = 661$), admixed American (AMR; $N = 347$), East Asian (EAS; $N = 504$), European (EUR; $N = 503$), and South Asian (SAS; $N = 489$). EUR includes five sub-populations: CEU, FIN, GBR, IBS, and TSI. In the analysis in which the HapMap CEU was used as a reference in HIBAG-prefit, we used the EUR data without CEU population ($N = 404$). The 1000 Genomes dataset includes 225,194 SNPs within the MHC region (chr6:28–35 Mb) obtained from sequencing and four-digit classical HLA types for *HLA-A, -B, -C, -DQB1,* and *-DRB1*. HLA genotypes were inferred from the NGS data using an in-silico typing software called PolyPheMe[42]. We downloaded the data from the 1000 Genomes project website[23]. For computational efficiency, we used 5539 SNPs that overlapped with the Immunochip.

*Korean panel*. The Korean panel[25] includes 5858 SNPs within the MHC region (chr6:25–35 Mb) genotyped using the Illumina's HumanOmniExpress platform and 4-digit classical HLA types for *HLA-A, -B, -C, -DPB1, -DQB1,* and *-DRB1*. HLA genes were genotyped using Roche's GS 454 sequencing at the Institute for Immunology and Infectious Diseases (Murdoch WA, Australia), followed by calling algorithms for HLA alleles that were accredited by the American Society for Histocompatibility and Immunogenetics[25]. We downloaded the SNP2HLA-formatted data for 413 individuals from the website described in Kim et al.[25].

*Chinese panel*. The Chinese panel[24] is the largest-to-date Asian reference panel for SNP2HLA. This dataset includes 29,948 SNPs within the MHC region (chr6:28–35 Mb) obtained from sequencing and four-digit classical HLA types for *HLA-A, -B, -C, -DPA1, -DPB1, -DQA1, -DQB1,* and *-DRB1*. This data also includes *HLA-DRB3, -DRB4,* and *-DRB5*, which were not used in our study. HLA genes were typed by the targeted NGS sequencing followed by in-silico typing software[50]. We obtained the SNP2HLA-formatted data for 10,689 individuals from the website described in Zhou et al.[24] We applied a quality control (QC) procedure to remove any individuals who did not have exactly two appearances of the "presence" in the binary markers at any gene, which left 9773 individuals.

*Pan-Asian panel*. Pan-Asian panel[51,52] includes 6173 SNPs within the MHC region (chr6:25–35 Mb) genotyped using the Illumina HumanHap1M and Affymetrix SNP 6.0 microarrays. The dataset includes four-digit classical HLA types for *HLA-A, -B, -C, -DPA1, -DPB1, -DQA1, -DQB1,* and *-DRB1*. HLA genes were typed using a sequence-based typing (SBT) method with taxonomy-based sequence analysis. The individuals consisted of the Singapore Chinese population ($N = 91$); pan-Asian datasets including 111 Chinese, 119 Indian, and 120 Malaysian subjects ($N = 350$) and the HapMap Phase II Japanese and Han Chinese (JPT + CHB) populations ($N = 89$). We downloaded the SNP2HLA-formatted data for 530 individuals from the SNP2HLA website[5]. The details of this dataset are described in Okada et al.[51] and Pillai et al.[52].

*1958 Birth Cohort panel*. The 1958 Birth Cohort (58BC) panel includes 6719 SNPs within the MHC region (chr6:29–35 Mb) genotyped using the Illumina Immunochip platform. This dataset includes four-digit classical HLA types for *HLA-A, -B, -C, -DRB1,* and *-DQB1*. We used the SNP2HLA-formatted data for 918 individuals that have been used in our previous study[5].

*Measuring imputation accuracy*. We measured the imputation accuracy as follows. In the test dataset, let $(A_1, A_2)$ be the true alleles obtained by HLA typing and let $(B_1, B_2)$ be the predicted alleles at an HLA gene of an individual. We calculated the matching score

$$\frac{\max(I(A_1 = B_1) + I(A_2 = B_2), I(A_1 = B_2) + I(A_2 = B_1))}{2} \quad (6)$$

for each individual, where I is an indicator function that is 1 if the alleles match and 0 otherwise. We averaged the scores over individuals to get the accuracy at an HLA gene. To get an overall accuracy, we averaged the accuracies over the genes.

This matching score can be interpreted as, given the two true alleles of an individual, how many of them were correctly predicted. At the same time, we can also interpret it as, given the two predicted alleles, how many were true alleles. Thus, this measure is symmetric and is related to both sensitivity and positive predictive value (PPV). However, when we measure the accuracy of a specific allele, sensitivity and PPV may differ. In this study, when we note the imputation accuracy of a specific allele, we used sensitivity: how many of the true alleles were predicted correctly.

There were special considerations in determining the allele matches. First, as described in the "Datasets" section above, HLA typing technologies used for various datasets were heterogeneous. In some datasets, typing was based on the sequence information in exon 2 and 3 for Class I and exon 2 for the Class II genes. Therefore, in our study, we decided to count a pair of alleles as a match if they were identical in amino acids at these exons. That is, we used the P-group designation for allele matching. Second, if the typed allele name has been deprecated in the recent version of IPD-IMGT/HLA database[27], for example, HLA-A*24:01, we excluded it from the calculation. Third, because distinguishing between HLA-DRB1*14:01 from HLA-DRB1*14:54 is a well-known conundrum[53], we counted them as a match. These rules were consistently applied to all methods for a fair comparison.

**Reporting summary**. Further information on research design is available in the Nature Research Reporting Summary linked to this article.

## Data availability

All of the datasets analyzed in this paper are public and published in other papers. The HapMap CEU and Pan-Asian panels are available at the SNP2HLA website, http://software.broadinstitute.org/mpg/snp2hla. The Korean panel is available at https://sites.google.com/site/scbaehanyang/hla_panel, and the Chinese panel is available at http://gigadb.org/dataset/100156. The T1DGC panel is available at https://repository.niddk.nih.gov/studies/t1dgc-special/?query=snp2hla upon request for research purposes. 1000 Genomes data are available at https://www.internationalgenome.org/category/population and the 1958 Birth Cohort data are available at https://ega-archive.org/datasets/EGAD00000000031.

## Code availability

The software is available at https://github.com/WansonChoi/CookHLA (DOI: 10.5281/zenodo.4294712) for noncommercial academic research use[54]. The code to reproduce the results of the paper is available upon request from the corresponding author.

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

## Acknowledgements
This work was supported by the National Research Foundation of Korea (NRF) (grant number 2019R1A2C2002608) and the Bio & Medical Technology Development Program of the NRF (grant number 2017M3A9B6061852) funded by the Korean government, Ministry of Science and ICT. This work was supported by the Creative-Pioneering Researchers Program funded by Seoul National University (SNU).

## Author contributions
B.H. conceived the project with contributions from S.R. and X.J. B.H. supervised the research. S.C. built the prototype method, and W.C. and H.L. optimized the algorithm. W.C. packaged the software. S.C., W.C., and H.L. performed the analysis. Y.L. and K.K. provided guidance in manuscript preparation. All authors wrote the paper.

## Competing interests
B.H. is the CTO of Genealogy, Inc. The remaining authors declare no competing interests.

## Additional information

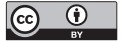

