## [Peer Review File · Nature Communications]

Reviewers' comments:

Reviewer #1 (Remarks to the Author):

In the manuscript, the authors proposed an in silico method "CookHLA" for HLA classical allele imputation using SNP data within the HLA region which are available in GWASs. CookHLA, like its predecessor "SNP2HLA", completely relies on the method and implementation of BEAGLE (one of the famous genome-wide genotype imputation softwares). To demonstrate the utilities and prediction accuracy of CookHLA, the SNP data and HLA classical genotypes from the Type 1 Diabetes Genetics Consortium (T1DGC) panel, HapMap CEU panel, 1000 Genomes panel, Chinese & Korean panels were used. The authors also show the application of HLA imputation and fine-mapping for rheumatoid arthritis, Type 1 diabetes and Crohn's disease. However, the manuscript is not of sufficient novelty, and the accuracy improvement from the customization in CookHLA is limited.

Comments:

1. The description of the statistical methods in BEAGLE should be provided, especially the algorithm details in BEAGLE_v3 (used by SNP2HLA) vs. BEAGLE_v4 (used by CookHLA). The BEAGLE_v4 imputation method assumes the input reference genotypes and target are phased (See "Material & Methods" in Brownings 2016). It is not clear how CookHLA deals with unphased genotypes (typical data in a GWAS), especially for the reference building. It is also not clear whether phased target SNPs or unphased target SNPs were used in the accuracy benchmark and computation time. Note that the downloaded dataset is the SNP2HLA-formatted data including phasing information. Imputation from unphased genotypes should take longer than phased data, even much longer.

2. Table 1, the fifth and last columns. The average accuracies based on BEAGLE_v4 (no genetic map, no local embedding of markers, 5th column) are just slight less than the accuracy of BEAGLE_v4 with adaptive map and local embedding (last column): 96.9% vs. 97.5%, and 97.1% vs. 97.5%. These results indicate the new customization in CookHLA is negligible. The major improvement is owing to the upgraded BEAGLE_v4 model and algorithm.

3. It is important to provide confident score for each imputed HLA classical genotypes, since the target sample does not almost have a good matching reference panel. In the manuscript, however there is no discussion on how to calculate the confident score of HLA classical alleles and how it impacts the accuracy and association.

4. L185, why only uses 200 individuals in genetic map estimation?

5. Datasets: the genotyping techniques should be mentioned, e.g., chip-based SNP platforms, NGS SNP data, Sequence-based typing (SBT) HLA typing, or NGS-based HLA typing. It would help to understand the heterogeneity of HLA genotypes due to genotyping techniques.

6. L67, it is not fair to limit the memory usage of BEAGLE_v3/SNP2HLA to max. 8GB, since machines with hundreds of GB memory are common in modern days. Multi-core functions are available in the HIBAG package, allowing for parallel calculation.

7. L108, it is not true that "these models were based on binary SNPs and were not directly suited for HLA genes that are in multi-allelic nature". BEAGLE_v3, v4 & v5 all support multi-allelic genotypes directly. The authors of BEAGLE have highlighted the multi-allelic ability in their BEAGLE_v5 paper (Brownings, 2018).

Reviewer #2 (Remarks to the Author):

In their manuscript, Cook et al. present CookHLA, a new method for statistical imputation of HLA alleles based on SNP genotype data.

CookHLA is the predecessor of SNP2HLA, a well-established HLA type imputation method that has been successfully applied in many genome-wide association studies.

The manuscript is well-written and mostly clear.

Strong points of the paper include a) the rare allele analysis; b) the nuanced analysis of the relative impact of the new version's different features on accuracy; c) analysing the differences between algorithms with respect to the conclusions one would draw in a fine-mapping analysis.

The validation results look good, but it would be good to include some experiments that help better understand the model's performance 1) on more diverse reference panels and 2) in the presence of more severe mismatches in terms of ancestry between the reference and the target cohorts.

Major points:

- The authors carry out four major validation experiments. They impute 1) from the T1DGC reference panel into HapMap; 2) from the T1DGC population into the European 1000G populations; 3) from T1DGC into T1DGC (cross-validation); 4) from a Chinese/Asian reference panel into a Korean population sample. The results of these experiments are instructive and they demonstrate the good performance of CookHLA; however, it would be important to a) carry out a test on a more diverse reference panel and b) evaluate what happens when the average ancestry of the reference panel significantly deviates from the average ancestry of the target panel. To address the first point, the authors could e.g. try to impute from the 1000 Genomes reference panel into the T1DGC panel; for the second point, they could try to e.g. impute from the combined Asian into the T1DGC panel. It would also be really cool to look at imputation performance in really diverse samples of African ancestry, but doing this properly based on the available panels may not be straightforward.

- It is unfortunate that a head-to-head comparison between CookHLA and HIBAG (head-to-head = training on exactly identical training data sets) could only be carried out for 3 of the 4 main experiments. I understand that the scalability of CookHLA is one of its strong points, and the cut-off of a maximum of 8 CPU months for model training seems sensible. However, given that the choices made by the authors mean that they end up with a single head-to-head experiment, it would be extra important to ensure that they correctly ran HIBAG - e.g. did the authors contact the HIBAG authors and ask for advice on improving runtimes? If the HIBAG runtimes are what they are, I would encourage the authors to include an additional validation experiment (perhaps in the Supplement, perhaps using some Asian data) in which the reference panel is chosen in a manner so as to allow for HIBAG model training from scratch.

- The authors write: "Since the gold standard HLA typing technologies commonly use the sequence information in the exon 2, 3, and 4 for the Class I and the exon 2 and 3 for the Class II genes.". This is not the case anymore - "gold standard HLA typing" would probably have to be understood as PacBio sequencing of long-range HLA amplicons these days, enabling full-resolution 4-field HLA typing (see e.g. the work by SGE Marsh et al., <https://pubmed.ncbi.nlm.nih.gov/30935664/>). The authors are right that clinical-grade HLA typing is typically based on only a subset of exons, but these are 2 and 3 for HLA class I (not 2, 3, and 4), and exon 2 for HLA class II (not 2 and 3). This resolution is called "G group resolution" or "3-field G group resolution" (http://hla.alleles.org/alleles/g_groups.html). The only reason to keep the author's definition of "allele identity" would be if all the datasets used in this study had indeed used a typing approach based on an extended exon set, but this is not the case; e.g. the T1DGC data were generated in the standard fashion by analysing exons 2 / 3 (class I) and exon 2 (class II). In summary, measuring everything at the level of G groups seems fine, but the definition of which exons go into the allele matching process should be corrected.

- Does CookHLA provide any imputation quality metrics? Is there a way to recognize low-quality imputations, or situations in which the ancestry of an input sample is not sufficiently well-represented in the reference panel? Based on the authors' description, I assume that some

measure of posterior probability for each allele would be available. It would be good to discuss the utility (or absence) of quality metrics and also show a calibration plot (if quality metrics are part of the output).

- I had to read the parts of the paper about "local embedding of prediction markers" multiple times to understand what is going on. It would be great if the authors could improve the description of this part. If I understand correctly, if there are x exons, the whole imputation process is repeated x times, and during the i -th iteration, the markers are placed in the middle of the i -th exon. After having run the imputation process x times, a consensus call is generated. Furthermore, if I understand correctly the only thing that changes between the i -th and the $(i+1)$ -th iteration is the genetic distance between the binary markers and the surrounding SNPs adjacent to the locus.

Minor points:

- In the rare allele analysis, the authors report "accuracy" on rare alleles. But what exactly is the employed accuracy measure here? Sensitivity? PPV? Total accuracy in individuals that carry at least one rare allele? It is important to distinguish between the probability to correctly impute a rare allele (sensitivity) and the probability that a predicted rare allele is really present (PPV).

- I think part of the HapMap HLA types also come from de Bakker et al. 2006 (<https://pubmed.ncbi.nlm.nih.gov/16998491/>)? If so it would be important to cite this paper.

- One of the major reference panels used in this paper, the T1DGC panel, is not publicly available anymore (or so the SNP2HLA website says). It seems possibly to apply for access to this reference panel, but only if carrying out research in the autoimmune disease / diabetes space. This should be mentioned somewhere. Also, it would be good if the authors extended the GitHub website of their method a bit, made it more user-friendly, and provided recommended reference panels for download (could e.g. be based on the 1000G? Or at least describe the steps to produce such a 1000G reference panel?).

- The authors describe HLA*IMP as using a haplotype graph - the original HLA*IMP uses the Li & Stephens model (Li and Stephens 2003). A haplotype graph, also inspired by the BEAGLE model, was only introduced in HLA*IMP:02.

All in all, very nice work!

Note from the editor: The report has been modified with additional comments from Reviewer 2 on the comments from Reviewer 1, see below. Questions from editorial are shown in asterisks (***) , with Reviewer 2's comments listed afterward.

1. Are the concerns around the contribution of BEAGLE to this algorithm validated?

I think it is fair to say that the BEAGLE model update accounts for the majority of the accuracy gain compared to SNP2HLA.

As the other reviewer points out, the observed differences within the "new BEAGLE model" class of validation experiments are relatively small, e.g. on the order of 0.5% when comparing the "new BEAGLE model + HapMap map + no local embedding" case (i.e, a reasonable baseline case) with "new BEAGLE model + adaptive map + local embedding" (i.e., the full CookHLA model).

Is this negligible or not? One could argue one way or the other, in particular as overall averages are not informative about accuracy for rare alleles or fine-mapping performance.

To assess the importance of the authors' modifications to the standard BEAGLE algorithm, they may incorporate the "baseline case" from above (new BEAGLE + HapMap map) into the rare allele analysis, and perhaps also into the fine-mapping experiment - if there are significant differences between their modified model and standard BEAGLE in these experiments, it would be relatively clear that the proposed modifications represent a genuine contribution.

2. Do the other reviewers' comments alter your stance on the conceptual advance and novelty of the study?

Yes, to the extent that upon reflection I agree that it would be important to better understand the contribution of the authors' modifications in comparison to the "vanilla" BEAGLE 4/5 model.

Also, the other reviewer correctly pointed out that BEAGLE supports multi-allelic markers. I had not appreciated this and think that specifying the HLA alleles as multi-allelic markers would probably be the most natural way to do HLA type imputation with BEAGLE - i.e. this should probably be included as a baseline case in the validation experiments.

3. Do you feel that the new analyses suggested by the other reviewers are likely to add value to the paper? Do you feel any requests are unfeasible or beyond the scope of the study?
All requests made by the other reviewer seem reasonable and feasible.

I am not familiar with the details of the implementation of the HIBAG model and its scaling behaviour, but I agree that it would be important to more comprehensively benchmark CookHLA against HiBAG. In my review, I suggested adding an additional validation experiment with a reference panel suitable for training the HIBAG model; alternatively, if the training data of the existing experiments could be processed by HIBAG by activating multithreading, this would even be preferable.

4. Do you agree/disagree with any of the concerns raised by the other reviewer?

I agree with almost all individual points raised by the other reviewer; with respect to overall significance/novelty, I would still be inclined to accept that the modifications proposed by the CookHLA authors lead to meaningful if small improvements in accuracy, but more clearly demonstrating this in the paper would be good.

Response to reviewers

[Summary of major changes]

We sincerely thank the reviewers for giving us critical and helpful comments. We substantially updated the manuscript to address these points. The major change was that now we include a comprehensive pairwise analysis using 10 HLA reference panels of different ethnicities and sizes. We evaluated the imputation performance of methods in every possible pair, by assigning one panel as reference and another as target. A total of 88 comparisons showed that our technical advances improved accuracy much compared to the base (vanilla) version of the method. In addition, this result emphasized that our method can be useful for HLA studies in underrepresented populations for which there is no large ethnically matched panel. We also addressed other points one by one, as described below.

Reviewers' comments:

Reviewer #1 (Remarks to the Author):

In the manuscript, the authors proposed an *in silico* method “CookHLA” for HLA classical allele imputation using SNP data within the HLA region which are available in GWASs. CookHLA, like its predecessor “SNP2HLA”, completely relies on the method and implementation of BEAGLE (one of the famous genome-wide genotype imputation softwares). To demonstrate the utilities and prediction accuracy of CookHLA, the SNP data and HLA classical genotypes from the Type 1 Diabetes Genetics Consortium (T1DGC) panel, HapMap CEU panel, 1000 Genomes panel, Chinese & Korean panels were used. The authors also show the application of HLA imputation and fine-mapping for rheumatoid arthritis, Type 1 diabetes and Crohn’s disease. However, the manuscript is not of sufficient novelty, and the accuracy improvement from the customization in CookHLA is limited.

Comments:

1. The description of the statistical methods in BEAGLE should be provided, especially the algorithm details in BEAGLE_v3 (used by SNP2HLA) vs. BEAGLE_v4 (used by CookHLA). The BEAGLE_v4 imputation method assumes the input reference genotypes and target are phased

(See “Material & Methods” in Brownings 2016). It is not clear how CookHLA deals with unphased genotypes (typical data in a GWAS), especially for the reference building. It is also not clear whether phased target SNPs or unphased target SNPs were used in the accuracy benchmark and computation time. Note that the downloaded dataset is the SNP2HLA-formatted data including phasing information. Imputation from unphased genotypes should take longer than phased data, even much longer.

We thank the reviewer for this point. The main difference of Beagle v3 and v4 is that Beagle v3 was using a haplotype model based on variable length hidden Markov model, while Beagle v4 uses the standard framework using the genetic map, along with some additional techniques to make imputation fast. We now added description explaining the difference between Beagle v3 and v4, as follows.

To impute the binary markers, CookHLA employed a state-of-the-art imputation model (Beagle v4 and v5) that can deal with millions of samples^{17,18}. Previously, SNP2HLA employed an old model (Beagle v3)²⁹. While the old model is based on the variable-length hidden Markov model based on haplotype graphs, the new model is based on the standard hidden Markov model that can incorporate the genetic distance as input. In particular, the new model improved both efficiency and accuracy by including only the common markers between target samples and reference samples in the model and using interpolation for the rest of the markers¹⁷, and by using a parsimonious state space that is a fraction of the size of the state space of the full model¹⁸.

As for the phasing, we found that our description was unclear. CookHLA uses the same phased reference data as SNP2HLA. These data can be built using MakeReference module of SNP2HLA. Although reference building performs phasing, once it is built, no phasing is required for existing panels. Similar to SNP2HLA, CookHLA uses unphased SNP data as input. Although Beagle v4 and v5 accept phased data, they can also accept unphased data, in which case Beagle internally phases them first. The measured computation time of CookHLA includes this internal phasing (otherwise it is much faster). In all our benchmarking, the input for all methods were unphased SNP data. We now made these points clearer as follows.

Similar to SNP2HLA, CookHLA uses phased reference data and unphased SNP input data. When unphased data is provided as input on the new Beagle models, the data is first phased internally and then imputed. Although the internal procedure has changed, its use from the user's point of view is the same as with SNP2HLA. CookHLA uses the same phased reference data format as SNP2HLA. Thus, any reference panel built for

SNP2HLA can be used for CookHLA. In addition, if anyone wants to build a new reference panel for CookHLA, the MakeReference module of SNP2HLA can be used.

2. Table 1, the fifth and last columns. The average accuracies based on BEAGLE_v4 (no genetic map, no local embedding of markers, 5th column) are just slight less than the accuracy of BEAGLE_v4 with adaptive map and local embedding (last column): 96.9% vs. 97.5%, and 97.1% vs. 97.5%. These results indicate the new customization in CookHLA is negligible. The major improvement is owing to the upgraded BEAGLE_v4 model and algorithm.

We thank the reviewer's point. Indeed, for that comparison, the contribution of BEAGLE was the largest. Since this was evaluation of a single instance, we expanded the analysis to a wider range of situations. We collected 10 HLA reference panels of different ethnicities and sizes. Then we tested every possible pair by assigning one as reference and another as target. This comprised a total of 88 comparisons. Many pairs presented situations that a large ethnically matched panel does not exist, in which case a panel of different ethnicity or small size should be used.

The results are summarized in the new **Table 1**, copied in the next page. In almost all pairs of comparison (except 3), our method showed superior accuracy compared to the bare version (CookHLA-HapMap; upgraded BEAGLE with HapMap genetic map).

Table 1. Pairwise accuracy benchmark using 10 reference panels. We collected 10 different reference panels of various populations and sizes, and tested each pair by assigning one as a reference and another as a target. The imputation accuracy was averaged over all available HLA genes for each pair. In each cell, the first number is the accuracy of the full version of CookHLA, while the number in the parentheses is the accuracy of CookHLA-HapMap (CookHLA with only imputation engine upgrade, where the HapMap genetic map is used). The grey cells present pairs with ethnically matched reference panels. The bold faced font shows the highest accuracy for each target sample.

		Reference panel									
		European			East Asian				South Asian	African	Ad Mixed American
		T1DGC	58BC	1000G EUR	Chinese	Korean	1000G EAS	Pan-Asian	1000G SAS	1000G AFR	1000G AMR
Sample size		5,225	918	503	9,773	413	504	530	489	661	347
Target sample	T1DGC		91.3 (91.2)	93.3 (92.0)	80.4 (71.1)	75.1 (58.6)	73.0 (69.6)	78.7 (73.3)	85.1 (75.8)	88.7 (76.5)	91.0 (83.0)
	58BC	97.5 (97.5)		96.8 (95.4)	89.1 (83.1)	81.5 (64.3)	75.6 (63.8)	74.1 (68.3)	80.6 (73.9)	92.0 (82.0)	94.9 (87.6)
	1000G EUR	98.3 (97.4)	95.8 (95.0)		89.4 (87.4)	81.0 (69.3)	82.8 (75.9)	76.0 (70.6)	85.3 (79.4)	91.1 (88.2)	95.6 (87.7)
	Chinese	80.8 (77.5)	69.0 (63.8)	73.2 (67.2)		90.5 (81.3)	92.3 (84.3)	89.2 (85.1)	80.3 (74.7)	58.4 (52.7)	68.4 (59.7)
	Korean	90.0 (87.8)	71.9 (68.5)	74.0 (65.9)	95.4 (95.2)		95.2 (86.0)	90.6 (85.8)	77.9 (70.4)	61.9 (55.0)	71.3 (58.5)
	1000G EAS	91.3 (80.6)	61.2 (48.3)	68.8 (62.7)	94.4 (94.4)	90.8 (82.0)		Sample overlap	78.9 (77.8)	57.9 (50.8)	63.8 (60.4)
	Pan-Asian	90.0 (89.9)	61.0 (61.1)	64.3 (62.9)	87.5 (89.7)	78.4 (73.4)	Sample overlap		83.8 (76.4)	58.1 (54.9)	59.9 (51.0)
	1000G SAS	94.0 (90.8)	78.2 (74.3)	82.9 (75.9)	88.5 (85.6)	79.5 (69.8)	87.0 (83.6)	88.7 (82.3)		74.9 (63.3)	79.4 (67.9)
	1000G AFR	89.3 (84.8)	68.9 (64.3)	74.8 (68.8)	59.1 (56.6)	45.0 (37.8)	50.2 (44.4)	47.6 (42.8)	53.3 (45.7)		85.9 (74.2)
	1000G AMR	89.3 (88.7)	78.9 (78.9)	82.7 (79.3)	77.0 (76.7)	67.0 (57.8)	68.7 (65.1)	64.8 (59.0)	72.4 (65.8)	80.0 (77.3)	

We plotted the accuracies in the following figure (now Figure 3):

Figure 3. Pairwise comparison across 10 different reference panels. We collected 10 reference panels of differing ethnicities and sizes. We then tested every possible pair by assigning one panel as reference and another as target, which comprised 88 tests after excluding overlapping sample pairs. We compared the full version of CookHLA to CookHLA-HapMap (upgraded engine, with HapMap genetic map). The dotted line indicates where the two methods’ imputation accuracies are equal.

This result shows that our technical advances (local exon embedding & adaptive genetic map) improved the accuracy meaningfully.

We note that this result is directly relevant to the diversity issue of the current HLA studies, and suggests that our method can enhance the HLA studies of underrepresented populations for which a large reference panel is not available. We summarized this whole analysis in the new result section named “Imputation using unmatched or small reference”.

3. It is important to provide confident score for each imputed HLA classical genotypes, since the target sample does not almost have a good matching reference panel. In the manuscript, however there is no discussion on how to calculate the confident score of HLA classical alleles and how it impacts the accuracy and association.

We thank the reviewer for this point. Indeed, a confidence score can be useful in applications, and it can be important to measure how the accuracy changes by the call rate. There was a difference in the scores in that HIBAG provides score per individual, while CookHLA provides score per possible allele. Therefore, we defined a new measure for individual and compared the accuracy versus call rate. It turned out that the relative performance of the methods did not change by the call rate, as shown in the supplementary figure below.

Supplementary Figure 4. Imputation accuracy versus call rate. For each method, we changed the threshold to call genotypes based on the confidence score provided by the method. We used the T1DGC subset (N=1,000) as reference and the rest (N=4,225) as target, similar to **Figure 2C**. The imputation accuracy was averaged over all available HLA genes. (In SNP2HLA, *HLA-DPA1* was excluded because there was a phenomenon that all 4 digit markers show 0 posterior probabilities in this dataset.)

We now describe this point in the new result section, as follows.

Call rate and accuracy

So far we have measured the accuracy of the methods assuming the best-guess imputed alleles. This corresponds to calling all alleles after imputation without considering the uncertainty. Sometimes, one may want to drop uncertain alleles and measure only the accuracy of the called alleles. Here we analyzed the relationship between accuracy and call rate in different methods. We used the T1DGC-cross experiment in **Figure 2C**. HIBAG-fit and HIBAG-prefit provide confidence score for each genotype (pair of alleles). In contrast, CookHLA provides a consensus posterior probability for each allele. We defined CookHLA's confidence score for a genotype as the posterior probability of the allele for a homozygous call and the sum of the posterior probabilities of the two alleles for a heterozygous call. SNP2HLA does not explicitly provide confidence score, but the posterior probability of each allele can be extracted from output. Thus, we can build a similar score to CookHLA. Because the definitions of the scores were different among methods, we varied the score threshold for each method separately to measure accuracy versus call rate. **Supplementary Figure 4** shows that, as expected, the accuracy increased when the call rate was reduced in all methods. Decreasing the call rate did not change the relative performance of the methods; CookHLA was superior to other methods regardless of the call rate.

4. L185, why only uses 200 individuals in genetic map estimation?

We thank the reviewer for this point. We chose this number because MACH algorithm suggests using 200-500 individuals, and we found that increasing this number did not change the imputation accuracy, despite a prolonged computation time. We now provide more description about this choice, along with the results in **Supplementary Table 1**.

Given the reference panel and the target samples to impute, we randomly select Q individuals from the reference and Q individuals from the target sample (**Figure 1C**). Using these $2 \times Q$ individuals, we use the Baum-Welch algorithm implemented in MACH v1.0³⁰ to obtain the transition probabilities. MACH tutorial recommends using 200~500 subsamples, so we chose $Q=100$ ($2 \times Q=200$). We found that increasing this number had little effect on performance of CookHLA (**Supplementary Table 1**). We convert the transition probabilities to genetic map using equation (2), which is used as input information to CookHLA.

5. Datasets: the genotyping techniques should be mentioned, e.g., chip-based SNP platforms, NGS SNP data, Sequence-based typing (SBT) HLA typing, or NGS-based HLA typing. It would help to understand the heterogeneity of HLA genotypes due to genotyping techniques.

We thank the reviewer for this point. We thoroughly searched literature to obtain this information and updated the manuscript to clarify the technology. We found information for all datasets except 58BC panel. It turned out that there was heterogeneity. In some panels, only exon 2&3 (or exon 2 in class II) was examined using SSOP technology. For this reason, as suggested by reviewer #2, we changed our accuracy measuring scheme to only look at those exons. That is, we shifted to using P-group resolution in measuring accuracy. We updated all numbers in the manuscript using this new scheme. Interestingly, this did not change the previous results much, where in most cases the accuracy was changed by only 0.1%.

6. L67, it is not fair to limit the memory usage of BEAGLE_v3/SNP2HLA to max. 8GB, since machines with hundreds of GB memory are common in modern days. Multi-core functions are available in the HIBAG package, allowing for parallel calculation.

We thank the reviewer for this point. For multi-core usage, we actually used more memory, since we used 8Gb per core. Thus, for 9 cores, we used 72Gb. We made this clearer in both text and supplementary table. The choice of 8Gb for single core usage was assuming a researcher without access to a big server. Still, someone may be able to use more memory. For that reason, we deleted the sentence about SNP2HLA not completing WTCCC imputation. To our experience, however, SNP2HLA's computation time was not much affected by memory as long as it finishes without crashing. Interestingly, increasing the memory for SNP2HLA sometimes increased computation time, counter-intuitively. We decided not to include those complicated phenomena, because CookHLA (Beagle v5, single core) is able to run within hours even with small memory (8Gb).

As for HIBAG, it uses little memory, so giving 8Gb to it was more than enough.

Thanks for informing the multi-core capability of HIBAG. For fitting T1DGC panel, we tried expanding the cores. Instead of using 8 cores (one core per gene), using multi-core function, we used 8 cores per gene (a total of 64 cores). Since one fifth of T1DGC

(N=1,000) was successfully fit in 27 days using 8 cores, we expected the multi-core fitting of the whole T1DGC would finish in one month ($27\text{days} \times (5225/1000) / 8\text{cores} = 18\text{ days}$). However, as of today (which passed one month), it still did not finish. Thus, it turns out that either (1) the fitting time increase super linearly with respect to sample size or (2) multi core function does not linearly decrease computation with respect to number of cores. We now describe this observation as follows.

For this benchmarking, we were not able to compare HIBAG-fit because the fitting of the T1DGC data did not finish in one month using 8 CPU cores (one core per each of 8 genes in T1DGC). By utilizing the multi-core functionality of HIBAG, we increased the number of CPUs to 8 cores per gene (total 64 cores), but it still did not finish in one month.

We found one previous study that also reported that HIBAG took prolonged time for fitting.

HLA-check: evaluating HLA data from SNP information

JeanMougin et al., BMC Bioinformatics 2017.

In this paper, the authors used the same T1DGC for SNP2HLA, but describe:

We also tried our method on HIBAG for two-field class I HLA genes, and it gave similar results and improvements (Table 5). Note that this is not directly comparable to the other imputation method since we used a pre-built reference file (this tool gives HLA imputation results much faster than other methods, **but the reference data for a given chip needs to be computed and is a very time-consuming process**), **so we could not control the reference panel or use the same as with SNP2HLA.**

We emailed the first author of HIBAG paper (the manager of the HIBAG github) to ask and discuss the computational time issue, but did not get an answer.

7. L108, it is not true that “these models were based on binary SNPs and were not directly suited for HLA genes that are in multi-allelic nature”. BEAGLE_v3, v4 & v5 all support multi-allelic genotypes directly. The authors of BEAGLE have highlighted the multi-allelic ability in their BEAGLE_v5 paper (Brownings, 2018).

We thank the reviewer for this point. Beagle software supports multi-alleles, but the functionality is only limited to A, T, G, C letters. If defined differently, it gives errors. Beagle v5 can accept any compositions of the letters. For example the allele name can be A, AA, AAA, AAAA, Thus, if we have 30 different alleles for an HLA gene, we can probably define a set of length 1 A to length 30 As, and use a pseudo-map to connect them to allele names. However, this was a too complicated change, so we decided to keep the binary scheme. We now modified the sentence to:

However, these models were mostly based on A/T/G/C SNPs and were not directly suited for HLA genes that are in multi-allelic nature.

We also describe further about our choice:

These new Beagle models had a functionality that can incorporate multiple alleles at a locus. However, they were implemented to only accept A, T, C, and G or compositions of these letters as alleles, and thus were not directly applicable to HLA nomenclature. Therefore, we chose to use the existing scheme that defines one binary marker per each HLA allele.

Reviewer #2 (Remarks to the Author):

In their manuscript, Cook et al. present CookHLA, a new method for statistical imputation of HLA alleles based on SNP genotype data.

CookHLA is the predecessor of SNP2HLA, a well-established HLA type imputation method that has been successfully applied in many genome-wide association studies.

The manuscript is well-written and mostly clear.

Strong points of the paper include a) the rare allele analysis; b) the nuanced analysis of the relative impact of the new version's different features on accuracy; c) analysing the differences between algorithms with respect to the conclusions one would draw in a fine-mapping analysis.

The validation results look good, but it would be good to include some experiments that help better understand the model's performance 1) on more diverse reference panels and 2) in the presence of more severe

mismatches in terms of ancestry between the reference and the target cohorts.

Major points:

- The authors carry out four major validation experiments. They impute 1) from the T1DGC reference panel into HapMap; 2) from the T1DGC population into the European 1000G populations; 3) from T1DGC into T1DGC (cross-validation); 4) from a Chinese/Asian reference panel into a Korean population sample. The results of these experiments are instructive and they demonstrate the good performance of CookHLA; however, it would be important to a) carry out a test on a more diverse reference panel and b) evaluate what happens when the average ancestry of the reference panel significantly deviates from the average ancestry of the target panel. To address the first point, the authors could e.g. try to impute from the 1000 Genomes reference panel into the T1DGC panel; for the second point, they could try to e.g. impute from the combined Asian into the T1DGC panel. It would also be really cool to look at imputation performance in really diverse samples of African ancestry, but doing this properly based on the available panels may not be straightforward.

We truly thank the reviewer for this point. We first thought that anyone would just use the ethnically matched panel, so designed the study that way. However, it is true that for studies of many small and underrepresented populations, it is likely there is no large ethnically matched panel and therefore a suboptimal panel must be used. To address this point, we performed a comprehensive analysis using 10 reference panels.

These 10 panels were of differing sizes and ethnicities. We tested every possible pair by assigning one as reference and another as target. This comprised a total of 88 comparisons. Many pairs presented situations that a large ethnically matched panel does not exist, in which case a panel of different ethnicity or small size should be used.

The results are summarized in the new **Table 1**, copied here. In almost all pairs of comparison (except 3), our method showed superior accuracy compared to the bare version (CookHLA-HapMap; upgraded BEAGLE with HapMap genetic map).

Table 1. Pairwise accuracy benchmark using 10 reference panels. We collected 10 different reference panels of various populations and sizes, and tested each pair by assigning one as a reference and another as a target. The imputation accuracy was averaged over all available HLA genes for each pair. In each cell, the first number is the accuracy of the full version of CookHLA, while the number in the parentheses is the accuracy of CookHLA-HapMap (CookHLA with only imputation engine upgrade, where the HapMap genetic map is used). The grey cells present pairs with ethnically matched reference panels. The bold faced font shows the highest accuracy for each target sample.

		Reference panel									
		European			East Asian				South Asian	African	Ad Mixed American
		T1DGC	58BC	1000G EUR	Chinese	Korean	1000G EAS	Pan-Asian	1000G SAS	1000G AFR	1000G AMR
Sample size		5,225	918	503	9,773	413	504	530	489	661	347
Target sample	T1DGC		91.3 (91.2)	93.3 (92.0)	80.4 (71.1)	75.1 (58.6)	73.0 (69.6)	78.7 (73.3)	85.1 (75.8)	88.7 (76.5)	91.0 (83.0)
	58BC	97.5 (97.5)		96.8 (95.4)	89.1 (83.1)	81.5 (64.3)	75.6 (63.8)	74.1 (68.3)	80.6 (73.9)	92.0 (82.0)	94.9 (87.6)
	1000G EUR	98.3 (97.4)	95.8 (95.0)		89.4 (87.4)	81.0 (69.3)	82.8 (75.9)	76.0 (70.6)	85.3 (79.4)	91.1 (88.2)	95.6 (87.7)
	Chinese	80.8 (77.5)	69.0 (63.8)	73.2 (67.2)		90.5 (81.3)	92.3 (84.3)	89.2 (85.1)	80.3 (74.7)	58.4 (52.7)	68.4 (59.7)
	Korean	90.0 (87.8)	71.9 (68.5)	74.0 (65.9)	95.4 (95.2)		95.2 (86.0)	90.6 (85.8)	77.9 (70.4)	61.9 (55.0)	71.3 (58.5)
	1000G EAS	91.3 (80.6)	61.2 (48.3)	68.8 (62.7)	94.4 (94.4)	90.8 (82.0)		Sample overlap	78.9 (77.8)	57.9 (50.8)	63.8 (60.4)
	Pan-Asian	90.0 (89.9)	61.0 (61.1)	64.3 (62.9)	87.5 (89.7)	78.4 (73.4)	Sample overlap		83.8 (76.4)	58.1 (54.9)	59.9 (51.0)
	1000G SAS	94.0 (90.8)	78.2 (74.3)	82.9 (75.9)	88.5 (85.6)	79.5 (69.8)	87.0 (83.6)	88.7 (82.3)		74.9 (63.3)	79.4 (67.9)
	1000G AFR	89.3 (84.8)	68.9 (64.3)	74.8 (68.8)	59.1 (56.6)	45.0 (37.8)	50.2 (44.4)	47.6 (42.8)	53.3 (45.7)		85.9 (74.2)
	1000G AMR	89.3 (88.7)	78.9 (78.9)	82.7 (79.3)	77.0 (76.7)	67.0 (57.8)	68.7 (65.1)	64.8 (59.0)	72.4 (65.8)	80.0 (77.3)	

We plotted the accuracies in the following figure (now **Figure 3**):

Figure 3. Pairwise comparison across 10 different reference panels. We collected 10 reference panels of differing ethnicities and sizes. We then tested every possible pair by assigning one panel as reference and another as target, which comprised 88 tests after excluding overlapping sample pairs. We compared the full version of CookHLA to CookHLA-HapMap (upgraded engine, with HapMap genetic map). The dotted line indicates where the two methods’ imputation accuracies are equal.

This result shows that our technical advances (local exon embedding & adaptive genetic map) improved the accuracy meaningfully.

Thus, we believe (and hope) that this analysis result has addressed both the reviewer’s one concern about the diverse population matching, as well as the reviewer’s another concern about our technical advances compared to the vanilla version. We summarized this whole analysis in the new result section named “Imputation using unmatched or small reference”.

- It is unfortunate that a head-to-head comparison between CookHLA and HIBAG (head-to-head = training on exactly identical training data sets) could only be carried out for 3 of the 4 main experiments. I understand that the scalability of CookHLA is one of its strong points, and the cut-off of a maximum of 8 CPU months for model training seems sensible. However, given that the choices made by the authors mean that they end up with a single head-to-head experiment, it would be extra important to ensure that they correctly ran HIBAG - e.g. did the authors contact the HIBAG authors and ask for advice on improving runtimes? If the HIBAG runtimes are what they are, I would encourage the authors to include an additional validation experiment (perhaps in the Supplement, perhaps using some Asian data) in which the reference panel is chosen in a manner so as to allow for HIBAG model training from scratch.

We thank the reviewer for this point. We double checked if we correctly ran HIBAG, and confirmed that we ran right. Since we had a success in fitting one instance (T1DGC subset N=1,000) in one month (8 cores, total 8 CPU months), it seems that it works as long as it finishes.

As the reviewer noted, we contacted the first author of HIBAG paper (the manager of the HIBAG github) to ask and discuss the computational time issue, but did not get an answer.

For fitting T1DGC panel, we tried expanding the cores. Instead of using 8 cores (one core per gene), using multi-core function, we used 8 cores per gene (a total of 64 cores). Since one fifth of T1DGC (N=1,000) was successfully fit in 27 days using 8 cores, we expected the multi-core fitting of the whole T1DGC would finish in one month ($27\text{days} \times (5225/1000) / 8\text{cores} = 18\text{days}$). This would have resolved the reviewer's request to add another HIBAG fit, since then we can measure HIBAG-fit's accuracy for both T1DGC→CEU (Figure 2a) and T1DGC→EUR (Figure 2b). However, as of today (which passed one month), it still did not finish. Thus, it turns out that either (1) the fitting time increase super linearly with respect to sample size or (2) multi core function does not linearly decrease computation with respect to number of cores. We now describe this point as follows.

For this benchmarking, we were not able to compare HIBAG-fit because the fitting of the T1DGC data did not finish in one month using 8 CPU cores (one core per each of 8 genes

in T1DGC). By utilizing the multi-core functionality of HIBAG, we increased the number of CPUs to 8 cores per gene (total 64 cores), but it still did not finish in one month.

Additionally, we found one previous study that also reported that HIBAG took prolonged time for fitting.

HLA-check: evaluating HLA data from SNP information

JeanMougin et al., BMC Bioinformatics 2017.

In this paper, the authors used the same T1DGC for SNP2HLA, but describe:

We also tried our method on HIBAG for two-field class I HLA genes, and it gave similar results and improvements (Table 5). Note that this is not directly comparable to the other imputation method since we used a pre-built reference file (this tool gives HLA imputation results much faster than other methods, **but the reference data for a given chip needs to be computed and is a very time-consuming process**), so we could not control the reference panel or use the same as with SNP2HLA.

As the reviewer suggested, it would be great if we could fit HIBAG for another dataset we used, but we expect that the fitting would take another ~8 CPU months per dataset. Because HIBAG multi-core function did not make the fitting of T1DGC feasible, we currently launched fitting of the 1000G EAS panel using HIBAG (which corresponds to a subset of experiment of **Supplementary Figure 1**), which did not finish. It would be great if we could wait very long to finish HIBAG fitting for all panels and compare comprehensively, but another doubt is that if anyone would really want to spend >8 CPU months to fit HIBAG to custom panel, in practice. As we describe in Discussion, we believe if one has a custom panel, running CookHLA may give more accurate results while finishing imputation in hours.

- The authors write: "Since the gold standard HLA typing technologies commonly use the sequence information in the exon 2, 3, and 4 for the Class I and the exon 2 and 3 for the Class II genes.". This is not the case anymore - "gold standard HLA typing" would probably have to be understood as PacBio sequencing of long-range HLA amplicons these days, enabling full-resolution 4-field HLA typing (see e.g. the work by SGE Marsh et al., <https://pubmed.ncbi.nlm.nih.gov/30935664/>). The

authors are right that clinical-grade HLA typing is typically based on only a subset of exons, but these are 2 and 3 for HLA class I (not 2, 3, and 4), and exon 2 for HLA class II (not 2 and 3). This resolution is called "G group resolution" or "3-field G group resolution" (http://hla.alleles.org/alleles/g_groups.html). The only reason to keep the author's definition of "allele identity" would be if all the datasets used in this study had indeed used a typing approach based on an extended exon set, but this is not the case; e.g. the T1DGC data were generated in the standard fashion by analysing exons 2 / 3 (class I) and exon 2 (class II). In summary, measuring everything at the level of G groups seems fine, but the definition of which exons go into the allele matching process should be corrected.

We thank the reviewer for this point. We removed the "gold standard" term from the manuscript.

We actually did not know that T1DGC was typed for exon 2 / 3 for class I and 2 for class II. We searched literature and confirmed this and cited the article. We also found that, by the paper suggested by the reviewer below, HapMap CEU was similarly typed for those exons.

To also address reviewer #1's comment, we searched literature for HLA typing technologies of the datasets we used and updated the manuscript with this information.

Also, we changed the accuracy measuring scheme to only look at exon 2/3 for class I and exon 2 for class II, as suggested by the reviewer. Since most of the panels argue that they are 4 digit typed, which defines unique amino acid sequences, we called two alleles matched if they were identical in amino acids in those regions. That is, we used the P group resolution. We changed all numbers in the manuscript using this scheme. Interestingly, this did not change the previous results much, where in most cases the accuracy was changed by only 0.1%.

We now updated the accuracy measurement part as follows:

There were special considerations in determining the allele matches. First, as described in the Datasets section above, HLA typing technologies used for various datasets were heterogeneous. In some datasets, typing was based on the sequence information in the exon 2 and 3 for the Class I and the exon 2 for the Class II genes. Therefore, in our study,

we decided to count a pair of alleles as a match if they were identical in amino acids at these exons. That is, we used the P-group designation for allele matching.

- Does CookHLA provide any imputation quality metrics? Is there a way to recognize low-quality imputations, or situations in which the ancestry of an input sample is not sufficiently well-represented in the reference panel? Based on the authors' description, I assume that some measure of posterior probability for each allele would be available. It would be good to discuss the utility (or absence) of quality metrics and also show a calibration plot (if quality metrics are part of the output).

We thank the reviewer for this suggestion. Our method gives posterior probability of each allele (normalized over all alleles). Also, we defined a confidence score of an individual as the sum of posterior probabilities of the two alleles (or a single allele for homozygous genotype). Based on this score, we compared the accuracy and call rate, as follows.

Supplementary Figure 4. Imputation accuracy versus call rate. For each method, we changed the threshold to call genotypes based on the confidence score provided by the method. We used the T1DGC subset (N=1,000) as reference and the rest (N=4,225) as target, similar to Figure 2C. The imputation accuracy was averaged over all available HLA genes. (In SNP2HLA, *HLA-DPA1* was excluded because there was a phenomenon that all 4 digit markers show 0 posterior probabilities in this dataset.)

We now describe this point in the new result section, as follows.

Call rate and accuracy

So far we have measured the accuracy of the methods assuming the best-guess imputed alleles. This corresponds to calling all alleles after imputation without considering the uncertainty. Sometimes, one may want to drop uncertain alleles and measure only the accuracy of the called alleles. Here we analyzed the relationship between accuracy and call rate in different methods. We used the T1DGC-cross experiment in **Figure 2C**. HIBAG-fit and HIBAG-prefit provide confidence score for each genotype (pair of alleles). In contrast, CookHLA provides a consensus posterior probability for each allele. We defined CookHLA's confidence score for a genotype as the posterior probability of the allele for a homozygous call and the sum of the posterior probabilities of the two alleles for a heterozygous call. SNP2HLA does not explicitly provide confidence score, so we excluded it from comparison. Because the definition of the score was different between HIBAG and CookHLA, we varied the score threshold for each method to measure accuracy versus call rate. **Supplementary Figure 4** shows that, as expected, the accuracy increased when the call rate was reduced in all methods. Decreasing the call rate did not change the relative performance of the methods; CookHLA was superior to HIBAG-fit and HiBAG-prefit regardless of the call rate.

We updated the implementation of our software on github to print out the confidence score of the individuals.

- I had to read the parts of the paper about "local embedding of prediction markers" multiple times to understand what is going on. It would be great if the authors could improve the description of this part. If I understand correctly, if there are x exons, the whole imputation process is repeated x times, and during the i -th iteration, the markers are placed in the middle of the i -th exon. After having run the imputation process x times, a consensus call is generated. Furthermore, if I understand correctly the only thing that changes between the i -th and the $(i+1)$ -th iteration is the genetic distance between the binary markers and the surrounding SNPs adjacent to the locus.

We thank the reviewer for this point. Indeed, we found that our description using 'multiple marker sets' was confusing. We found that it's actually not duplication or

multiple marker sets—instead, it's repetition using different embedding as the reviewer pointed out. We removed the term “multiple marker set” from the manuscript and updated the description as follows.

To address this challenge, we have developed a strategy that repeats imputation multiple times while embedding each marker set locally in the middle position of each polymorphic exon (**Figure 1B**). Specifically, CookHLA repeats imputation for exon 2, 3, and 4 in Class I and exon 2 and 3 in Class II with an embedded marker set. To make final consensus calls over multiple imputation results, we combine the posterior probabilities of the binary markers as follows.

Based on the reviewer's comment, we also noted that our description of genetic map was unclear. The truth is that the genetic map for SNPs is learned from data, and fixed for all process. In each embedding, we embed markers and define the genetic map of those. We clarified this as follows:

We note that adaptive learning builds the genetic map of intergenic SNPs and not that of the binary markers, since the binary markers are missing in the target samples. When we embed the binary markers in the exon, we augment the map by defining the binary markers' genetic map. We first calculate the center position of the exon in terms of the genetic distance and place the markers at that position while specifying a very small genetic distance (10^{-12} cM) between the binary markers. This serves to suppress an undesired transition between the binary markers in the hidden Markov model, since binary markers are by definition exclusive (a chromosome cannot have two alleles).

Minor points:

- In the rare allele analysis, the authors report "accuracy" on rare alleles. But what exactly is the employed accuracy measure here? Sensitivity? PPV? Total accuracy in individuals that carry at least one rare allele? It is important to distinguish between the probability to correctly impute a rare allele (sensitivity) and the probability that a predicted rare allele is really present (PPV).

We thank the reviewer for this point. Our overall accuracy measure can both be interpreted as sensitivity and PPV. However, for allele-specific analysis, it was not clear. We were using sensitivity (number of correctly predicted alleles divided by the true allele count). We made this point clearer as follows.

This matching score can be interpreted as, given the two true alleles of an individual, how many of them were correctly predicted. At the same time, we can also interpret it as, given the two predicted alleles, how many were true alleles. Thus, this measure is symmetric and is related to both sensitivity and positive predictive value (PPV). However, when we measure the accuracy of a specific allele, sensitivity and PPV may differ. In this study, when we note the imputation accuracy of a specific allele, we used sensitivity: how many of the true alleles were predicted correctly.

- I think part of the HapMap HLA types also come from de Bakker et al. 2006 (<https://pubmed.ncbi.nlm.nih.gov/16998491/>)? If so it would be important to cite this paper.

We thank the reviewer for this information. We updated the description about the HapMap CEU dataset, and cited this paper.

- One of the major reference panels used in this paper, the T1DGC panel, is not publicly available anymore (or so the SNP2HLA website says). It seems possibly to apply for access to this reference panel, but only if carrying out research in the autoimmune disease / diabetes space. This should be mentioned somewhere. Also, it would be good if the authors extended the GitHub website of their method a bit, made it more user-friendly, and provided recommended reference panels for download (could e.g. be based on the 1000G? Or at least describe the steps to produce such a 1000G reference panel?).

We thank the reviewer for this point. We updated the description of T1DGC to include the following sentence,

This panel is available for research purposes per request.

We extensively updated our website to be more user friendly. Also, we uploaded the five super population datasets of 1000G panels (in SNP2HLA panel format, directly usable for CookHLA or SNP2HLA). Anyone can download these data to replicate our results.

- The authors describe HLA*IMP as using a haplotype graph - the original HLA*IMP uses the Li & Stephens model (Li and Stephens 2003). A haplotype graph, also inspired by the BEAGLE model, was only introduced in HLA*IMP:02.

We thank the reviewer for this point. In fact, the cited paper comparing different methods was based on specifically HLA*IMP:02. Thus, we updated the text HLA*IMP to be HLA*IMP:02.

All in all, very nice work!

Note from the editor: The report has been modified with additional comments from Reviewer 2 on the comments from Reviewer 1, see below. Questions from editorial are shown in asterisks (***) , with Reviewer 2's comments listed afterward.

1. Are the concerns around the contribution of BEAGLE to this algorithm validated?

I think it is fair to say that the BEAGLE model update accounts for the majority of the accuracy gain compared to SNP2HLA.

As the other reviewer points out, the observed differences within the "new BEAGLE model" class of validation experiments are relatively small, e.g. on the order of 0.5% when comparing the "new BEAGLE model + HapMap map + no local embedding" case (i.e, a reasonable baseline case) with "new BEAGLE model + adaptive map + local embedding" (i.e., the full CookHLA model).

Is this negligible or not? One could argue one way or the other, in particular as overall averages are not informative about accuracy for rare alleles or fine-mapping performance.

To assess the importance of the authors' modifications to the standard BEAGLE algorithm, they may incorporate the "baseline case" from above (new BEAGLE + HapMap map) into the rare allele analysis, and perhaps also into the fine-mapping experiment - if there are significant differences between their modified model and standard BEAGLE in these experiments, it would be relatively clear that the proposed modifications represent a genuine contribution.

We thank the reviewer for this point. As the reviewer pointed out, there was a major concern if our technical advances over the vanilla version is meaningful. To address this point, as described above, we added a comprehensive analysis of pairwise comparison using 10 reference panels. The data showed that our full version (full CookHLA) substantially increased accuracy over the vanilla version (CookHLA-HapMap). The increase was particularly large in comparisons where the reference panel's ethnicity was not matched to target's, suggesting the utility of our method for underrepresented populations.

As for the reviewer's request to look at the difference of CookHLA and CookHLA-HapMap for rare allele analysis, we performed this comparison. It turned out that, for T1DGC cross experiment that was originally performed in the rare allele analysis section, the two versions were almost the same (except DRB1), suggesting the superior accuracy on rare alleles of CookHLA was driven by the upgraded Beagle engine for this instance. (See **Supplementary Figure 3A** copied below)

However, when we repeated the analysis on a different dataset (Reference: 1000G EAS, Target: Korean), the rare allele accuracy showed meaningful difference between CookHLA and CookHLA-HapMap (**Supplementary Figure 3B** copied below).

We described this point as below:

We then wanted to compare different versions of CookHLA, as defined in the previous section. **Supplementary Figure 3A** shows that CookHLA-vanilla and CookHLA-HapMap achieved similar accuracy to the full CookHLA, except in (0.1~1.0%) bin *HLA-DRB1*, showing that the increased accuracy in rare alleles was mainly due to the imputation engine upgrade in this setting. However, when we repeated the similar analysis assuming the 1000 Genomes EAS data as reference and the Korean data as target, the full CookHLA was much more accurate than CookHLA-vanilla or CookHLA-HapMap in lower frequency bins (**Supplementary Figure 3B**). Thus, for the rare allele analysis, the contributions of the components in CookHLA showed similar trend as were for the overall accuracy; for the situation where a large ethnically-matched reference panel was used, the upgraded engine contributed the most, and for the situation where a small panel was used, the exon embedding and adaptive map strategies contributed significantly.

In sum, we expect that whether it is a situation that a large ethnically matched panel is available or in it is a situation that such a panel is unavailable, our method can provide good performance in imputing rare alleles. In the former, the contribution of BEAGLE would be large, and in the former, the contribution of our two strategies would be significant.

2. Do the other reviewers' comments alter your stance on the conceptual advance and novelty of the study?

Yes, to the extent that upon reflection I agree that it would be important to better understand the contribution of the authors' modifications in comparison to the "vanilla" BEAGLE 4/5 model.

Also, the other reviewer correctly pointed out that BEAGLE supports multi-allelic markers. I had not appreciated this and think that specifying the HLA alleles as multi-allelic markers would probably be the most natural way to do HLA type imputation with BEAGLE - i.e. this should probably be included as a baseline case in the validation experiments.

We thank the reviewer for this point. We compared our method to the vanilla version in the comprehensive pairwise comparison of 10 panels, which showed that our technical advances had meaningful improvement.

As for the multi-allele function, although Beagle accepts multi-alleles, it can only accept A/T/G/C or a composition of these letters. It would be possible to define a complicated mapping between arbitrarily generated A/T/G/C sequences and the HLA alleles, but that would be a complicated work-around, so we decided to stick to the binary marker scheme currently, which still gave us superior accuracy and efficiency to other methods.

3. Do you feel that the new analyses suggested by the other reviewers are likely to add value to the paper? Do you feel any requests are unfeasible or beyond the scope of the study?
All requests made by the other reviewer seem reasonable and feasible.

I am not familiar with the details of the implementation of the HIBAG model and its scaling behaviour, but I agree that it would be important to more comprehensively benchmark CookHLA against HiBAG. In my review, I suggested adding an additional validation experiment with a reference panel suitable for training the HIBAG model; alternatively, if the training data of the existing experiments could be processed by HIBAG by activating multithreading, this would even be preferable.

We thank the reviewer for this point. As described above, we tried multi core functionality of HIBAG, but it surprisingly still did not finish fitting T1DGC panel in 64 CPU months. (By linear calculation, it should have finished in 17 days, but it did not). If it had finished, we would have had HIBAG-fit for Figure 2a and 2b at the same time, but it did not work.

We are still currently fitting T1DGC, consuming 64 cores in the server continuously. This is a large consumption in our small lab, because we have other projects to run, but this one process is consuming a significant amount of our lab server for a month. Considering that, we really doubt if any individual researcher would practically be able to perform this fitting, unless affiliated to an institution with an access to large clusters.

We later launched fitting of the 1000G EAS panel, using 5 additional CPUs for 5 genes, but not sure when it will finish. We hope that it will finish in 5 CPU months.

Although we could not add one more reference fitted by HIBAG as requested, we added an additional analysis comparing CookHLA and HIBAG while changing the call rate, which is presented in **Supplementary Figure 4**.

4. Do you agree/disagree with any of the concerns raised by the other reviewer?

I agree with almost all individual points raised by the other reviewer; with respect to overall significance/novelty, I would still be inclined to accept that the modifications proposed by the CookHLA authors lead to meaningful if small improvements in accuracy, but more clearly demonstrating this in the paper would be good.

We truly thank the two reviewers' comments; We feel that addressing these points as much as possible have strengthened and clarified our paper much.

Reviewer #1 (Remarks to the Author):

In general, the new table (Table 1) in the revised manuscript addresses my concern on how the customization in CookHLA improves the Beagle_v4/v5 imputation. It seems that a custom genetic map contributes to the accuracy imputation, while the contribution of BEAGLE is the largest. However, this new table does not fully support the conclusion that CookHLA is a highly accurate HLA imputation method.

Comments:

1. It is strongly suggested to cite the well-known paper of "Li and Stephens model" (GENETICS Dec 1, 2003 vol. 165 no. 4 2213-2233), since most of coalescent-theory-based imputation methods (including Beagle v4 & v5) are derived from "Li & Stephens model". That is the major difference between Beagle_v3 and Beagle_v4, so that a genetic map could be incorporated into Beagle_v4.
2. Since CookHLA uses phased reference data, the uncertainty or phasing quality in the reference data can be an important factor in accuracy when the CookHLA imputation is used, especially for the training samples from diverse populations and having no family or trio data. It is worth mentioning the limit of building reference panel in the discussion section.
3. For the replication of a research study, the authors should provide a complete description on the HIBAG training parameters, since the authors report long computation time for building HIBAG models. For example, the version of the HIBAG package, the R version, how many individual classifiers in the HIBAG ensemble model (N=100?) for each gene, and the flanking region size (500K on each side?) if it was used. With large computer clusters, building HIBAG models using T1DGC (5K) and Chinese Han (10K) samples are not very time-consuming, since the HIBAG algorithm is highly parallelizable.
4. In addition, the authors mentioned both Beagle_v4 and Beagle_v5 in the manuscript, it is not clear which version of Beagle was actually used for Table 1 and Figure 2.
5. Supplementary Table 6. Computation time of different methods. Did the authors add the computation time for adaptively learning the genetic map?
6. In the revised manuscript, the accuracy is shifted to the P-coded resolution. According to the definition of P-group, exon 2 & 3 are used for class I genes, and only exon 2 is used for class II genes. Hence, for class II genes (e.g., HLA-DRB1), there is no local embedding needed because there is one exon only. The number of beagle runs in L187 & L188 may need to be updated. The importance of local embedding is still not very clear.

Reviewer #2 (Remarks to the Author):

Cook et al. have now submitted a revised version of their manuscript presenting CookHLA, a new HLA type imputation method.

The revision convincingly addresses concerns raised during the previous round of reviews about the relative contributions of different components of their method, in particular when compared to using the new Beagle version in isolation. Specifically, Table 1 shows that the "full CookHLA" outperforms a "reduced CookHLA" instantiation of the algorithm that uses the new Beagle model and a HapMap-based genetic map.

I also really like the experimental setup underlying Table 1, i.e. the systematic exploration of differing degrees of mismatch between reference and inference panels. This is an important experiment of relevance to HLA type imputation in general, and in some of the considered scenarios performance is better than I would have expected (for any HLA type imputation method).

The revised version also fully addresses the comments I had raised on the previous version. I only have two remaining minor remarks:

- For multi-allelic imputation, it does really not matter from a computational point of view whether the imputed alleles are called A*02:01, A*03:01, etc., or 1, 2, 3 etc., or A, AA, AAA etc. The authors remark e.g. "However, these models were mostly based on A/T/G/C SNPs and were not directly suited for HLA genes" - I don't think that the A/C/G/T point is really relevant and I would simply state that multi-allelic encoding of HLA types was not explored.

- The authors have clarified that their "Accuracy" metric for rare alleles is equivalent to sensitivity (e.g. in Figure 4). Sensitivity is obviously important, but also reporting a metric that penalizes potential over-imputation of rare alleles would also be good. I would therefore recommend also reporting PPV.

Response to reviewers

Reviewer #1 (Remarks to the Author):

In general, the new table (Table 1) in the revised manuscript addresses my concern on how the customization in CookHLA improves the Beagle_v4/v5 imputation. It seems that a custom genetic map contributes to the accuracy imputation, while the contribution of BEAGLE is the largest. However, this new table does not fully support the conclusion that CookHLA is a highly accurate HLA imputation method.

Comments:

1. It is strongly suggested to cite the well-known paper of “Li and Stephens model” (GENETICS Dec 1, 2003 vol. 165 no. 4 2213-2233), since most of coalescent-theory-based imputation methods (including Beagle v4 & v5) are derived from “Li & Stephens model”. That is the major difference between Beagle_v3 and Beagle_v4, so that a genetic map could be incorporated into Beagle_v4.

We thank the reviewer for this suggestion. We agree that the major contribution of all these methods stemmed from the ground-breaking PHASE model. We cited this paper when we describe our method in the Overview in Results. (Due to the editorial policy, we have moved the Results first, and made a short Overview section.)

Overview of the method

We developed an accurate HLA imputation method, CookHLA. Similar to its predecessor, SNP2HLA⁵, CookHLA translates the multi-allelic HLA information into a set of binary markers so that it can utilize an existing imputation algorithm (**Figure 1a**). CookHLA employs the state-of-the-art imputation engine¹⁸ that is superior to the one employed by SNP2HLA. In addition, CookHLA uses two strategies to maximize imputation accuracy. As depicted in **Figure 1b**, CookHLA repeats imputation while locally embedding prediction markers in each of the polymorphic exons and performs consensus calls. This strategy enables the binary markers to capture the local information contained in each exon more effectively compared to the naïve strategy of SNP2HLA that puts markers only in the center position of the gene. Next, CookHLA

adaptively learns the genetic map from the data (**Figure 1c**). Many imputation models, including the one we use, are based on the Li and Stephens model²⁷ that assumes each target individual as a mosaic of reference samples. In this model, the genetic map is used to determine how long a mosaic block stretches before switching to another block. Since the MHC region is notorious for the complex genetic structure that differs across populations²⁸, imputation can improve by learning the population-specific and data-specific genetic map from data instead of using the widely used HapMap map obtained from averaging several populations²⁰.

2. Since CookHLA uses phased reference data, the uncertainty or phasing quality in the reference data can be an important factor in accuracy when the CookHLA imputation is used, especially for the training samples from diverse populations and having no family or trio data. It is worth mentioning the limit of building reference panel in the discussion section.

We thank the reviewer for this point. Indeed, there can be phasing errors. Nevertheless, we found that for imputation of individual alleles, potential phasing errors are not much a problem because our accuracy tests were all based on reference panels with imperfect phasing. However, the phasing errors can be problematic for haplotype analysis. We discussed this point as follows, in Discussion.

Since CookHLA uses phased reference data, the phasing quality in the reference data can be an important factor in accuracy. Many of the existing reference panels available for CookHLA are not trio data, and therefore there may be phasing errors. As for the imputation accuracy of the individual alleles, we observed that the potential phasing errors are not much problematic, because our accuracy tests were all based on reference panels with imperfect phasing. However, if we perform haplotype analysis, the phasing errors may affect the results more severely. In the future, it would be ideal to build population-specific reference panels by minimizing phasing errors with trio or family data.

3. For the replication of a research study, the authors should provide a complete description on the HIBAG training parameters, since the authors report long computation time for building HIBAG models. For example, the version of the HIBAG package, the R version, how many individual classifiers in the HIBAG ensemble model (N=100?) for each gene, and the flanking region size (500K on each side?) if it was used. With large computer clusters, building HIBAG models using T1DGC (5K) and Chinese Han (10K) samples are not very time-consuming, since the HIBAG algorithm is highly parallelizable.

We thank the reviewer for this point. Yes, we have used $N=100$ (the default number of classifiers). We used the default setting described in the last section of https://zhengxwen.github.io/HIBAG/hibag_index.html. We found that some of the examples in this page included the flanking region option, as suggested by the reviewer. So, we tried to confine SNPs to 500kb regions to each side of the gene. Then we ran HIBAG again using 64 cores (8 cores per gene). However, even using this flanking region option, the fitting of any of the genes did not finish in two weeks using Intel Xeon CPU 2.1Ghz (a total of 32 CPU months). We described the details in Methods, as follows.

HIBAG also allows users to fit a new model using custom reference data, which should be done per each gene. We found that this fitting can take a long time. We tried to fit 8 classical genes in T1DGC data using 8 computer CPU cores (Number of Classifiers = 100), but the fitting did not finish in one month (8 CPU months). We then confined SNPs to flanking regions (500kb on each side of a gene) and tried 64 CPU cores (8 cores per gene), but the fitting did not finish in 2 weeks (32 CPU months). We used Intel Xeon 2.1Ghz CPU for this benchmark.

4. In addition, the authors mentioned both `Beagle_v4` and `Beagle_v5` in the manuscript, it is not clear which version of Beagle was actually used for Table 1 and Figure 2.

We thank the reviewer for this point. Most of our analyses were done using v4, since we began this project a long time ago and v4 was the only available option. We wanted to use v4 for all analyses for consistency, but for the pairwise analysis of 10 reference panels, the use of v5 was inevitable because we had to finish all ~90 comparisons within the short revision period. In Methods, we already described that we used v4 except otherwise stated (which is the pairwise analysis in Table 1), but it would have been better to clarify this at multiple locations of the draft.

Now, in Methods, we write

We found that Beagle v4 and Beagle v5 gave almost identical results in our application. In this study, we used Beagle v4 for most of the analyses, unless otherwise stated.

In Results, where we describe pairwise analysis (Table 1), we write

To further increase efficiency, we used Beagle v5 for this analysis instead of v4.

In caption of Table 1, we write

For computational efficiency, we used Beagle v5 instead of v4.

5. Supplementary Table 6. Computation time of different methods. Did the authors add the computation time for adaptively learning the genetic map?

We thank the reviewer for this point. Indeed, we found that the current table did not include the time of estimating adaptive genetic map (AGM). We added the time in the Supplementary Table. We found that the additional time did not much affect the practicability of the method, since it was only about 0.5 hour.

We updated the text in Results as well, as follows.

CookHLA with Beagle 4 was slower than SNP2HLA for Task 1 (19.4 hours) and faster than SNP2HLA for Task 2 (29.1 hours). Since CookHLA performs ensemble learning from 9 runs, it can be easily parallelized with multiple CPUs. With 9 CPUs, CookHLA became much faster, taking 2.6 hours for Task 1 and 4.5 hours for Task 2. CookHLA with Beagle 5 was even faster. With a single CPU, it took 1.4 hours for both Task 1 and 2, and with 9 CPUs, it only took about 10 minutes for both tasks. **Note that for CookHLA, an additional time of about 0.5 hour was required for preparing adaptive genetic map.**

6. In the revised manuscript, the accuracy is shifted to the P-coded resolution. According to the definition of P-group, exon 2 & 3 are used for class I genes, and only exon 2 is used for class II genes. Hence, for class II genes (e.g., HLA-DRB1), there is no local embedding needed because there is one exon only. The number of beagle runs in L187 & L188 may need to be updated. The importance of local embedding is still not very clear.

We thank the reviewer for this point. As the reviewer pointed out, in this revision, we evaluated the accuracy of the imputation based on alleles in P-group, which is exon 2&3-based for class I and exon2-based for class II. We made this choice in our manuscript because for many datasets we analyzed in the paper, the alleles were only typed based on these exons. Thus, the gold standard answers we had already included ambiguity. In that sense, it was not surprising that local embedding did not much help in this context.

We checked if we change our method scheme to only look at exon 2&3 in class I and exon 2 in class II, what will happen. We observed that only a slight decrease in accuracy in analyses in Figure 2a, where the accuracy was slightly decreased from 97.6% to 97.4%. Thus, as shown in **Supplementary Table 2**, local embedding is useful when used alone, but not as much useful when used with adaptive genetic map, although it still helps a bit.

Nevertheless, the inevitable situation that we chose P-group resolution for accuracy evaluation was only forced by the fact that our data were ambiguous. In the future, NGS-based typing will allow us to build many 6-digit reference panels. In such situations, it will be interesting to see if the local embedding will help more.

We discussed these points in Discussion, as follows.

Our method introduced two strategies to improve imputation, namely local embedding and adaptive genetic map. When we compared the contributions of the two strategies, we found that the contribution of the adaptive genetic map was greater (**Supplementary Table 2**). One reason for this might be because for many datasets we used, only exon 2 and 3 for class I and exon 2 for class II were typed. Thus, the alleles had ambiguity. Since the class II genes were typed based on a single exon, it is not surprising that local embedding did not much help. Since NGS-based typing is becoming prevalent, we expect that in the future, reference datasets with 6-digit allele information will be available. Then, we will be able to evaluate the actual gain of local embedding using these datasets.

Reviewer #2 (Remarks to the Author):

Cook et al. have now submitted a revised version of their manuscript presenting CookHLA, a new HLA type imputation method.

The revision convincingly addresses concerns raised during the previous round of reviews about the relative contributions of different components of their method, in particular when compared to using the new Beagle version in isolation. Specifically, Table 1 shows that the "full CookHLA" outperforms a "reduced CookHLA" instantiation of the algorithm that uses the new Beagle model and a HapMap-based genetic map.

I also really like the experimental setup underlying Table 1, i.e. the systematic exploration of differing degrees of mismatch between reference and inference panels. This is an important experiment of relevance to HLA type imputation in general, and in some of the considered scenarios performance is better than I would have expected (for any HLA type imputation method).

Thanks a lot for the comments. We truly appreciate the reviewer's helpful critiques.

The revised version also fully addresses the comments I had raised on the previous version. I only have two remaining minor remarks:

- For multi-allelic imputation, it does really not matter from a computational point of view whether the imputed alleles are called A*02:01, A*03:01, etc., or 1, 2, 3 etc., or A, AA, AAA etc. The authors remark e.g. "However, these models were mostly based on A/T/G/C SNPs and were not directly suited for HLA genes" - I don't think that the A/C/G/T point is really relevant and I would simply state that multi-allelic encoding of HLA types was not explored.

We thank the reviewer for this point. Truly, it would be possible to model multi-alleles using that scheme. We now have changed the sentence to

In order to utilize these models in HLA imputation, SNP2HLA coded the presence and absence of each allele as a binary marker (**Figure 1A**).

And added the following sentences,

Note that since a recent imputation framework supports modeling of multi-alleles¹⁸, an alternative approach would have been directly modeling alleles instead of using binary markers. In the current study, this alternative option was not explored.

- The authors have clarified that their "Accuracy" metric for rare alleles is equivalent to sensitivity (e.g. in Figure 4). Sensitivity is obviously important, but also reporting a metric that penalizes potential over-imputation of rare alleles would also be good. I would therefore recommend also reporting PPV.

We thank the reviewer for this recommendation. It is possible that CookHLA shows superior performance just because it tends to call rare alleles. To check this, we

calculated PPV as suggested. **Supplementary Figure 4a** shows that, CookHLA has similar PPV as other methods. Finally, we calculated F1-score, which can be considered a balanced measure to account for both sensitivity and PPV. We found that in terms of F1-score, CookHLA was superior to other methods (**Supplementary Figure 4b**, shown below).

We discussed this in Results, as follows.

Since we defined accuracy as sensitivity, one possible concern in this analysis can be whether our method is overly imputing rare alleles. To examine this, in the cross-validation of T1DGC panel, we measured positive predicted value (PPV) for each allele. **Supplementary Figure 4a** shows that CookHLA has similar PPV as other methods. Finally, we calculated F1-score, the harmonic mean of sensitivity and PPV. When averaged over the genes, CookHLA showed superior F1-score than other methods (**Supplementary Figure 4b**). In the lowest frequency bin, (0.1~0.5%), CookHLA achieved the highest F1-score (0.88) while the second best method HIBAG-fit achieved 0.78. In the second lowest frequency bin (0.5~1.0%), CookHLA achieved the highest F1-score (0.89) while the second best method SNP2HLA achieved 0.83. In the third lowest frequency bin (1.0~5.0%), again, CookHLA achieved the highest F1-score (0.96) while the second best method HIBAG-fit achieved 0.94.